# Learning to operate an imagined speech Brain-Computer Interface involves the spatial and frequency tuning of neural activity
Kinkini Bhadra [1], Anne-Lise Giraud[1,2,3] & Silvia Marchesotti [1,3] ✉

Brain-Computer Interfaces (BCI) will revolutionize the way people with severe impairment of speech production can communicate. While current efforts focus on training classifiers on vast amounts of neurophysiological signals to decode imagined speech, much less attention has been given to users' ability to adapt their neural activity to improve BCI-control. To address whether BCI-control improves with training and characterize the underlying neural dynamics, we trained 15 healthy participants to operate a binary BCI system based on electroencephalography (EEG) signals through syllable imagery for five consecutive days. Despite considerable interindividual variability in performance and learning, a significant improvement in BCI-control was globally observed. Using a control experiment, we show that a continuous feedback about the decoded activity is necessary for learning to occur. Performance improvement was associated with a broad EEG power increase in frontal theta activity and focal enhancement in temporal low-gamma activity, showing that learning to operate an imagined-speech BCI involves dynamic changes in neural features at different spectral scales. These findings demonstrate that combining machine and human learning is a successful strategy to enhance BCI controllability.

Neurological disorders of language such as aphasia, amyotrophic lateral sclerosis, and locked-in syndrome can disrupt natural speech dramatically impacting the quality of life for both patients and caregivers[1,2]. A promising approach to restore language communication is to decode imagined speech directly from neurophysiological signals and translate them into text, synthesized speech or even moving avatars through a brain-computer interface (BCI). This approach has raised two important challenges: how the machine can decode neural signals, and how the patient can optimize its interaction with the decoder. For the latter, providing a feedback to the user in real-time is crucial.

Recent years have seen great advances in the field of speech-BCIs, most often through the decoding of attempted speech from intracranial electrophysiological recordings[3–9], which have led to impressive decoding speeds reaching about 78 words per minute[4]. Such an approach, however, is unlikely to be suitable for disorders of language where speech production areas are damaged, such as in post-stroke expressive aphasia. A more appropriate BCI

for these disorders would require decoding imagined, rather than attempted speech. Also termed covert or inner speech, imagined speech consists of the internal production of speech without self-generated audible output[10,11], thus without the involvement of the musculoskeletal system. Depending on the brain damage location, different imagined speech strategies can be considered, from kinesthetic to abstract phonological ones. Although previous studies have characterized the neural correlates of imagined speech[12], mostly in comparison with overt speech[13–18], only a handful of BCI studies have attempted to decode imagined speech in real-time, with promising but often limited effectiveness[19–22]. This is due to different challenges and limitations primarily pertaining to the weakness of imagined speech signals as compared to overt speech[13,14,16,17,23], the difficulty in precisely identifying the onset of speech imagery[23], inter-individual differences in the ability to control the BCI[24,25], and the technique employed to record brain activity.

The state-of-the-art approach to experimentally address imagined speech decoding is to exploit intracranial recordings such as electrocorticography

[1]Department of Basic Neurosciences, Faculty of Medicine, University of Geneva, Geneva, Switzerland. [2]Université Paris Cité, Institut Pasteur, AP-HP, Inserm, Fondation Pour l'Audition, Institut de l'Audition, IHU reConnect, Paris, France. [3]These authors contributed equally: Anne-Lise Giraud, Silvia Marchesotti. ✉e-mail: silvia.marchesotti@unige.ch

(ECoG) and stereotactic EEG (sEEG), which allow neural sampling from key language regions with higher spatial resolution and to use higher-frequency neural activity than with surface recordings. Exploiting these experimental advantages, imagined speech decoding for BCI-control has been first attempted using ECoG to decode imagined phoneme pronunciation versus rest from the perisylvian area[19]. Although decoding accuracy was highly above chance, this study did not provide evidence that the method could be used to discriminate between two imagined speech units in real-time. More than a decade later, sEEG was used to synthesize imagined speech in real-time into continuous acoustic feedback from high-gamma activity in the frontal cortex and motor areas[21]. Although the reconstructed speech was unintelligible and less accurate for imagined than overt speech, this study was the first proof of concept that imagined speech could be used for naturalistic communication with a speech neuroprosthesis. More recently, impressive real-time control (up to 91%) was achieved by decoding eight imagined words from single-neuron activity in the supramarginal gyrus[22], highlighting the superior effectiveness of decoding speech from individual neurons. Yet, using intracortical recordings for speech-BCIs remains a clinical and ethical challenge, owing to the high risk of clinical complications (loss of contacts, infection[26]) potentially requiring explantation and the loss of the new communication means, a dramatic outcome for the patient[27]. Much research and clinical efforts are still required to optimize the success of future speech-BCIs.

Capitalizing on its far greater ease of use, several studies have employed surface EEG for decoding offline (i.e., open-loop) a wide variety of imagined speech units such as phonemes, syllables, and words, most often in binary classification paradigms[10,28] (see for reviews[10,29,30]). However, nearly all studies address imagined speech decoding from an engineering perspective, their main goal being the optimization of current classifiers to boost decoding accuracy (see[31] for a review of classification methods). Despite the great amount of data and the possibility of applying computationally demanding decoders, offline classification accuracy from pre-recorded datasets remains below 80% when discriminating between two imagined speech units and around 60% for a three-class problem[28,32–35].

In the single BCI study that used EEG for online (i.e., closed-loop, real-time) speech imagery decoding[20], performance remained below 70% in discriminating between "yes" and "no". Interestingly, however, this study pointed out important inter-individual differences in BCI-control, with accuracies varying between 53.75% and 95%[20]. The variability in control abilities is well known in motor-imagery EEG-BCIs, in which up to 50% of participants are unable to achieve above chance BCI-control[24,36]. Given the lack of speech-imagery EEG-BCI studies and the fact that invasive-BCI studies are mostly single-case, it remains to be assessed whether speech-BCI skills can improve with training. In the present study, we addressed speech-BCI controllability from a neurophysiological rather than neuroengineering perspective. We investigated whether BCI-control performance can be trained, and identified the neural and behavioral mechanisms underpinning the acquisition of these new skills. We designed a closed-loop BCI system based on EEG signals to decode in real-time the imagery of two syllables /fɔ/ and /gi/, chosen for their contrasted phonetic features, and trained 15 healthy participants to control the BCI for 5 consecutive days. We addressed the importance of feedback accuracy in BCI control learning by comparing these data with those obtained in a group of 10 healthy participants who trained with a discontinuous real-time feedback. This study thus targets both the variability and the dynamic range that can be achieved via training a whole brain EEG speech-imagery BCI.

## Materials and methods
### Participants
Fifteen healthy participants (5 females, mean age 23.9 years, SD ± 2.3, range 19–29) took part in this study which was approved by the local Ethics Committee (Commission Cantonale d'Éthique de la Recherche, project 2022-00451) and was performed in accordance with the Declaration of Helsinki. All ethical regulations relevant to human research participants were followed. All participants provided written informed consent and received financial compensation for their participation. All participants were right-handed.

### Experimental paradigm and syllables imagery
Participants took part in the study daily for 5 consecutive days, at the same time of the day. To avoid a potential effect of circadian fluctuations on the participants' performance, each of them began the training at the same time each day. Each session lasted approximately 2.5 h, amounting to a total duration of 12–13 h of experimental time per participant. The experiment took place in an optically, acoustically, and electrically shielded room.

On each day, participants performed a mental chronometry task followed by a BCI-control session, both involving the imagery of two syllables /fɔ/ or /gi/, chosen for their contrasted phonetic features regarding consonant manner (fricative vs plosive), place of articulation (labiodental vs velar), vowel place (mid back vs high front) and rounding (rounded vs unrounded). As different neural responses are associated with these distinct phonetic features[17,37–39], we expected to maximize the discriminability between the EEG signals associated with the imagery of each syllable. Participants were asked to focus on the kinesthetic sensation they would experience if they pronounced the syllable aloud. As the long-term goal of speech-BCI is to provide a means of communication for individuals who have lost the ability to speak, and consistent with the latest works of speech-BCI[4,52], participants were instructed to focus on how they would articulate speech rather than how speech would sound or look like in writing. Using imagined articulating speech, we expected to obtain a consistent neural response across the entire group and thus get reliable EEG analyses. Another technical advantage of exploring first this strategy is that kinesthetic imagery recruits more superficial brain areas than imagined speech perception thus more accessible with surface EEG[40].

At the end of the last and 5th day of training, participants were asked to report the strategy they used during the BCI-control session (see Supplementary Table 1 for individual reports).

### Mental chronometry
The mental chronometry test is a well-known experimental approach to empirically evaluate motor imagery skills (see for instance[41]) that has previously been used to evaluate motor imagery abilities for BCI-control[25]. According to this previous literature, the temporal congruency between the time required to perform the motor imagery and its actual execution indicates good imagery abilities (and vice-versa). Here we applied this methodology for the first time to the speech domain to probe a possible relationship between interindividual variability in speech imagery timing (acutely and across training days) and BCI performance, and to gain insights into the pace used to perform the imagery.

To do so, we asked our participants to either repeat aloud (i.e., overt) or imagine pronouncing (i.e., covert) five times one of two syllables used for the BCI control. Participants were instructed to verbally report the moment in which they began and completed the task by saying respectively "start" and "stop". The time between these two verbal indications was measured with a chronometer by the experimenter. There were a total of four experimental conditions with modality (speak/imagine) and syllable (/fɔ/ or /gi/) as factors, each of which was repeated 10 times. Participants were instructed to keep a constant rhythm for repeating the syllables throughout the trials.

### EEG acquisition and BCI loop
**EEG recording.** Neural data were recorded using a 64-channel ANT Neuro system (*eego mylab*, ANT Neuro, Hengelo, Netherlands) at a sampling rate of 512 Hz using electrode AFz as ground and CPz as reference. Channels' impedance was kept below 20 kΩ throughout the experiment. Electromyography signals (EMG) were recorded from the right side of the participant's face to measure potential articulatory muscles' activation despite our explicit instructions to avoid any movement[42]. The zygomaticus major and the orbicularis oris were targeted as these are most prominently involved in the place of articulation of the two syllables (respectively for /gi/ and /fɔ/[42]) and the right side was determined by the participants' handedness (all right-handed), typically matching the dominant side of the face, and therefore tends to exhibit more pronounced movements during speech production[43,44]. EEG and

EMG data were acquired using Lab Streaming Layer (LSL, https://github.com/sccn/labstreaminglayer).

During the EEG recording and while operating the BCI, participants sat comfortably on a chair in front of a computer screen while keeping their hands on their thighs and were instructed to avoid any physical movement.

**BCI loop.** The EEG-BCI loop was developed using an adapted version of the framework *Neurodecode* (Fondation Campus Biotech Geneva, https://github.com/fcbg-hnp/NeuroDecode), already used in previous BCI studies[45–47]. On each training day, the BCI-control included two sessions, an *offline* session in which data were recorded for the classifier's calibration and an *online* part where participants controlled a visual feedback in real-time. Therefore, the classifier was different on each experimental day. Participants were asked to use the same imagery strategy throughout the entire duration of the BCI training (see *Syllable imagery during BCI-training* section).

**Syllable imagery during BCI-training.** Participants were instructed to imagine repeating each of the two syllables (such as "/fɔ/-/fɔ/-/fɔ/…" or "/gi/-/gi/-/gi/-…" using the cognitive strategy described in the above section "*Experimental paradigm and syllables imagery*") while keeping a constant pace. Unlike the mental chronometry test, they were not instructed to imagine a specific number of syllable repetitions, and the experimenter made no reference to this previous cognitive task. Participants were explicitly told to avoid any movements during imagery, especially those involving the face, and not to mouth nor whisper. They were informed that the muscle activity of the face was being monitored with EMG electrodes throughout the entire experiment to control for such movements.

**Offline session and classifier calibration.** EEG data were acquired while participants performed syllable imagery without receiving any real-time feedback (*offline* runs) and subsequently used to calibrate the classifier. Each *offline* trial began with a text indicating the trial number (1 s), followed by a fixation cross (2 s), and a written cue indicating which of the two syllables participants had to imagine pronouncing (2 s). After the cue disappeared, an empty battery appeared on the screen, which then progressively filled for 5 s (Fig. 1a). Participants were instructed to start imagining pronouncing the syllable immediately after the battery appeared on the screen and stop when the battery was filled. They were explicitly told that the battery filling was independent of their brain signals. At the end of the 5 s imagery period, the battery was displayed as it appeared at the last filling level, for 2 additional seconds, with the tip of the battery turning yellow to indicate the participant to stop imagining. Participants had 5 s to rest while the instruction 'Rest' was displayed on the screen. There were a total of 40 trials per syllable, arranged in 4 blocks each consisting of 10 trials per syllable, with short breaks in between blocks. The *offline* session lasted approximately 25–30 min.

*Offline* data was then used to calibrate the decoder. Features were extracted by computing the power spectral density (PSD) of the EEG signal from 1 to 70 Hz (with a 2 Hz resolution) using a sliding window of 500 ms and 20 ms overlap. The PSD was calculated for each EEG channel excluding the three electrodes placed over the mastoid region bilaterally and the reference channel, leading to 61 channels. Therefore, there were a total of 2135 features (61 channels and 35 frequencies), each of which consisted of a channel-frequency pair. These features were fed to a random forest (RF) algorithm to extract the classifier parameters (i.e., the covariance matrix). This nonlinear classifier has already been proven effective in previous two-class BCI studies[48] and is known to be robust to overfitting[49]. The RF classifier assigns a weight (expressed in percentage) to each feature, indicating its relative contribution to the classification. An 8-fold cross-validation (CV) was performed to test the model validity and calculate the *offline* CV accuracy.

**Online BCI-control.** During the *online* part of the experiment, the RF classifier was applied in real-time to the EEG data to decode which of the two syllables the participant was imagining, and accordingly, provided a continuous real-time feedback to the user to inform them about their neural performance. Trials were the same as during the *offline* part and participants were instructed to fill the battery by performing the same imagery task as before, keeping the same pace. The mapping of the decoder output to the battery feedback at each time sample was done in such a way that if the probability output by the classifier changed in the direction of the cued syllable, the battery's filling would increase, and it would decrease if it didn't. The real-time control went on until the battery was full or until a 5 s timeout. The delay between the recorded data and the feedback presentation was on average 100 ms. As for the *offline* session, there were 40 trials per syllable, divided into 4 blocks.

To boost participants' motivation and keep them engaged in the task throughout the entire training period, we provided monetary bonuses based on performance. Participants received an additional 10 CHF for each experimental day in which their CV accuracy during the *offline* session was above chance (50%) and was higher than on the previous day.

**Experiment with discontinuous real-time feedback.** Ten healthy participants (5 females, mean age 24.9, SD ± 4.38 years, range 18–35, different individuals from those included in the main study) took part in a separate experiment in which the real-time feedback was experimentally altered. As in the main experiment, participants trained to control a BCI for 5 consecutive days, through the imagery of the same two syllables (/fɔ/ and /gi/). The sequence of the different experimental parts, imagery instructions, EEG acquisition, and BCI-system were the same as the one described in the previous paragraph, except that the real-time feedback was discontinuous, i.e., not systematically related to the classifier output and displayed only positive changes (see Supplementary Fig. 1 and Supplementary Methods for more details about the paradigm). Unlike in the main experiment, participants were presented with an auditory cue similar to the sound of a metronome, imposing a pace for the syllable repetition arbitrarily set at 1.4 Hz.

**Data analyses**

**BCI Performance and CV accuracy.** Participants' BCI-control performance was calculated by considering, for each trial, the percentage of the classifier's outputs that corresponded to the cued syllable. To probe for an increase in BCI-control performance, we performed a planned contrast analysis, considering the average performance during each training day for each participant, and testing for a linear increase or decrease from day 1 to 5 (numeric contrast using as weights −2, −1, 0, 1, and 2, respectively for day 1 to 5). This set of contrasts was tested in a linear mixed model (LMM), with participants as a random factor. The same statistical approach was used to probe changes across days in other dependent variables (e.g., features weight and power modulation evolution), and it is hereinafter referred to as "*LMM with planned contrast*".

We investigated whether the learning dynamics (i.e., the evolution of performance across the 5 training days) were related to the individual's ability to control the BCI. To do so, we fitted, separately for each participant, a linear model considering the average performance on each day and extracted the individual learning slope. We then computed the Pearson correlation between the average BCI-control performance across the whole training period and the learning slope.

Additionally, we tested for differences in performance between the two syllables (2-tailed paired t-test) and between blocks (one-way repeated measures ANOVA with *Block* number as within-participant factor, and 2-tailed paired t-tests for post hoc comparisons).

To further assess potential learning mechanisms across training, we considered the CV accuracy of the classifier obtained using the *offline* data (same as "*classifier calibration*" section) and applied the same method to the *online* session data. This approach pools together all data from an individual

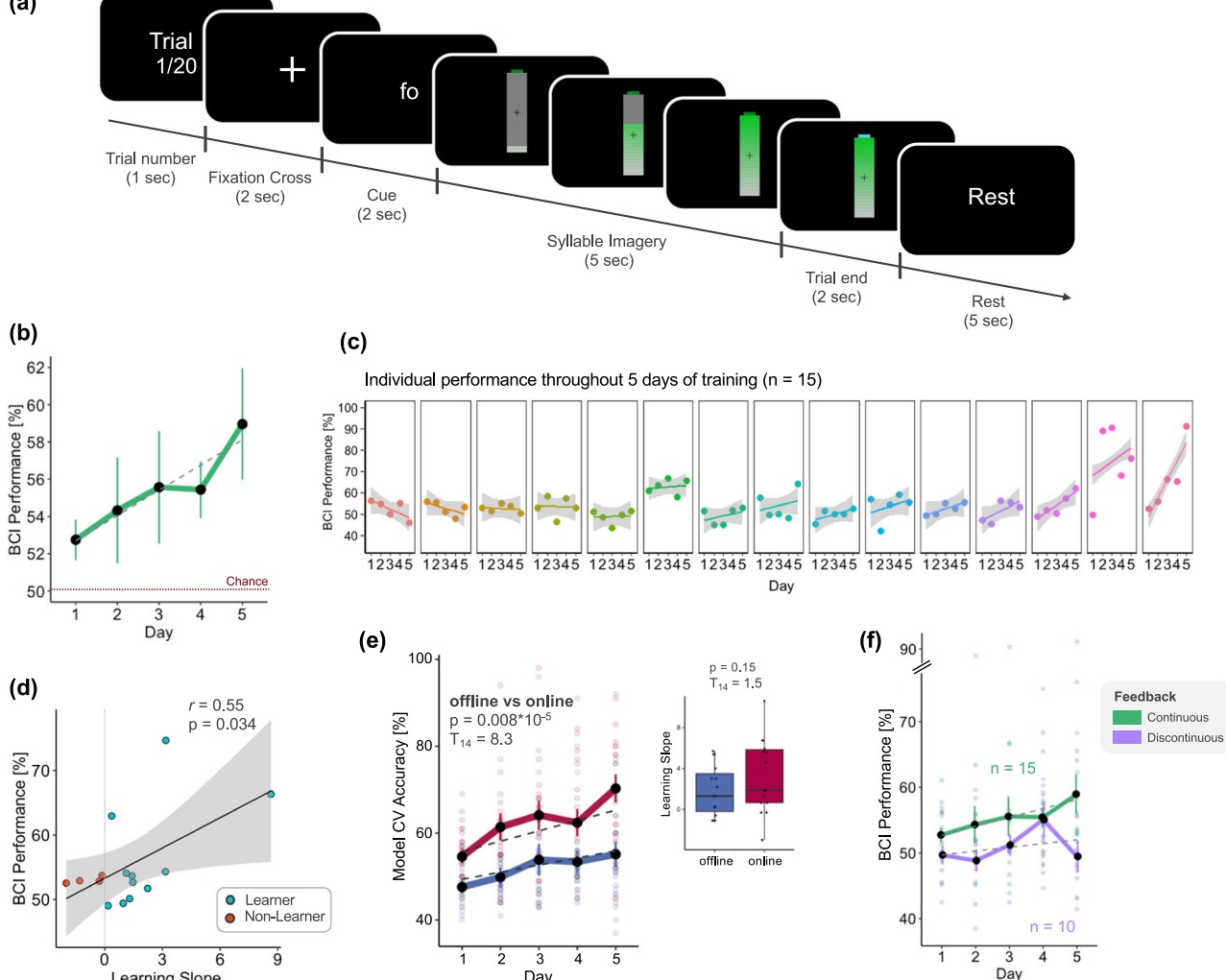

**Fig. 1 | Experimental paradigm, *online* BCI performance, and decoding accuracy.**
**a** Experiment Paradigm. Each trial began with the presentation of the trial number, followed by a fixation cross. The last second of the fixation cross was considered as a baseline for the EEG analysis. Next, participants were instructed to imagine pronouncing either the syllable /fɔ/ or /gi/ as indicated by a text on the screen. They had to continue performing the imagery for 5 s in the *offline* session, or until the battery was filled during the real-time control (*online* session). The end of the imagery period was signaled by the tip of the full battery turning yellow. **b** Average BCI *online* performance (%) over the 5 training days. BCI performance is computed, separately for each trial, as the percentage of classifier outputs in accordance with the cued syllable. Dots represent the average performance across participants with error bars indicating the standard error of the mean, and the significant linear regression (dashed line). **c** *Online* BCI performance (%) for each participant was ordered according to the value of the learning slope (from lowest to highest). Dots represent the individual average BCI-control performance for each training day, with the corresponding regression line.

**d** Correlation between individual learning slopes and average BCI performance across the 5 training days. Positive and negative learning slopes are plotted in cyan and orange, respectively for learners and non-learners. **e** Model cross-validation (CV) accuracy was obtained by computing the classifier considering the entire dataset from the *offline* (blue) and *online* (red) sessions on each day (left). The box plot (right) shows the learning slopes obtained by fitting a linear model per participant and session (*offline* in blue and *online* in red) using CV accuracies on each training day. There was no significant difference between the two sessions. Boxes represent the interquartile range (IQR), with the horizontal line indicating the median, and whiskers extending to data points that are within 1.5× the IQR from the upper and lower quartile. Individual points represent data from a single participant, and gray lines connect data points from the same participant. **f** Impact of discontinuous feedback on learning dynamics. Comparison of average BCI *online* performance (%) with the continuous (green, *n* = 15, same as **b** vs discontinuous real-time feedback (purple, *n* = 10). Error bars in (**b**, **e**, **f**) indicate the standard error of the mean.

session (*offline* or *online*) and thus differs from the method used to compute the *BCI-control performance*, which is based on individual samples at the single-trial level. We tested for a linear increase in CV accuracy using the *LMM with planned contrast*, separately for the *offline* and *online* sessions. We compared the CV accuracy between the two sessions, averaged across days, with a 2-tailed paired t-test.

To test for differences in learning between *offline* and *online* sessions, we computed the training slope considering the CV accuracy across the 5 days for each participant and each session (*offline*/*online*) using the same method as for the *BCI-control performance*. We then assessed differences in CV-accuracy slope between *offline* and *online* sessions by performing a 2-tailed paired t-test.

## Impact of the discontinuous real-time feedback on learning

To address whether the accuracy and consistency between the decoded brain patterns and the real-time feedback were a critical factor in BCI-control learning, we performed a subset of the analyses carried out for the main experiment, on the data acquired using the discontinuous feedback (behavioral and classifier's data). First, we evaluated the presence of the learning pattern in BCI-control performance (%) across the 5 days of training as observed in the main experiment, using the approach defined above as "*LMM with planned contrast*".

Next, we compared CV accuracies between the two groups by running a linear mixed model with fixed factors the planned contrast modeling a linear increase ("*LMM with planned contrast*"), the *Group* (continuous/

discontinuous feedback), and the *Session* (*offline/online*). We performed post-hoc comparisons by performing a two-tailed paired t-test for within-group comparisons and a two-tailed unpaired t-test for between-group comparisons.

## Classifier features

Next, we investigated which brain regions and frequency bands contribute most to BCI-control and studied changes in decoding patterns across training days. For this, we considered the feature weights of the classifier used during real-time BCI-control and computed based on the *offline* session data. Each feature refers to a specific channel-frequency pair leading to a total of 61 (channels) x 35 (frequencies) features. The weight of each feature is expressed as a percentage, where higher values indicate a stronger contribution in discriminating between the two syllables.

First, we addressed whether better BCI-control is associated with higher features' weights, reflecting a better discriminability between the two classes. To do so, we considered, separately for each participant, the sum of the weights across the first 200 features (i.e., irrespective of frequency or channel location, ranked according to their weight). This sub-sampling was necessary since the sum of all features' weights would have led to the same value of 100% across all participants. This subset size was chosen based on the cumulative sum of the first 200 features' weights exceeding on average 50% and on its standard deviation across training days increasing up to the 170th ranking place (Supplementary Fig. 2a). This shows that only a part of higher ranking features is most prominently involved in training. We performed a Pearson correlation coefficient between the feature's weight sum, averaged across days, and the average individual BCI-control performance (obtained as described in the *BCI Performance and CV accuracy* section).

Next, we investigated the topography of the most discriminant features, as well as their frequency distribution. We considered the average weight over the 5 training days (1) separately for each individual frequency disregarding to which electrode the features belonged and (2) across the scalp separately for each frequency value and frequency band, to obtain a topographical representation of the feature weights.

The frequency distribution as well as the topography for the most discriminat frequency interval was also computed for the dataset acquired using the discontinuous feedback.

## Evolution of features over training

Subsequently, we quantified the evolution of the features' weight over the five training days. First, to assess global changes in the weights across training, we considered the weights' sum of the first 200 features for each training day and ran the "*LMM with planned contrast*" analysis, testing for a linear change in the weight with training. A positive relationship between BCI-control performance and weight over the 5 days would reflect a behavioral improvement.

Next, we explored changes due to training more specifically at the level of the decoding frequencies and brain regions. We first inspected changes by visualizing feature pairs as individual elements in a 2D map, with frequency values on the x-axis and individual channels on the y-axis.

We quantified changes over time separately in the frequency and spatial domain (i.e., at the topographic level). To limit the number of multiple comparisons, we guided the spatial analyses by the results obtained in the frequency domain.

To identify frequency-wise changes to the discrimination between the two syllables, we considered, separately for each individual frequency value the average weight across all features and performed a "*LMM with planned contrast*". We assessed the statistical significance of the linear change across the 5 days of training with 1000 permutations, assigning randomly the day to which data belonged, within participants. Based on these results, we defined frequency intervals, over which performing the analysis in the spatial domain. We then considered the average weight across all features and ran a "*LMM with planned contrast*" for each electrode and each interval. The statistical significance was assessed with 1000 permutations, again by randomizing the factor *Day* on a single-participant basis.

Next, we evaluated the relationship between changes in BCI-control performance across days and changes in the features space. To quantify the global feature changes in a single index per participant, we considered the Euclidean distance between the feature weights of two consecutive days according to the following formula (Eq. 1):

$$dist(DayN+1, DayN) = \sqrt{\sum_{iFeature}\left(DayN+1_{iFeature} - DayN_{iFeature}\right)^2}$$

(1)

where N indicates the experimental Day and iFeature a Frequency-Channel pair (Fig. 2f). The four distance matrices (one for each couple of consecutive days) were then averaged to obtain an individual index per participant, referred to hereafter as the *global index*. Last, we correlated the *global index* with the average BCI performance computed considering all training days.

## EEG data preprocessing and analysis

EEG data recorded during the *online* session were preprocessed using Fieldtrip[50] and the Semi-Automatic Selection of Independent Component Analysis (SASICA)[51] toolboxes within the MATLAB environment (version R2018b; The MathWorks, Natick, MA, USA). The data were first filtered using a zero-phase Butterworth bandpass filter with cutoff frequencies of 1 and 70 Hz. They were then divided into epochs of 12 s centered around the syllable imagery onset, including 4 s pre-stimulus (during the fixation cross and the cue presentation) and 8 s post-imagery onset (5 s of *online* BCI control and 3 s of rest). Noisy channels and epochs were removed via visual inspection, after which the data were re-referenced to the common average (which also served the purpose of retrieving data from the reference channel). Principal Component Analysis (PCA) was used to identify and remove ocular and muscular artifacts. The choice of the components to be removed was guided by different metrics such as autocorrelation, focal trial activity, and dipole fit residual, computed through the SASICA toolbox. Last, noisy channels that were initially removed were added back to the dataset by interpolation.

## Power changes during BCI-control

To investigate oscillatory modulations associated with speech imagery and BCI control, we computed the power change for each individual frequency and each channel during the entire trial with respect to the baseline activity in the −3 to −2 s pre-imagery onset (i.e., during the fixation cross presentation). For this, we used Morlet wavelets to decompose the preprocessed EEG time series in the time-frequency domain. The power values were baseline-normalized at the single trial level and separately for each frequency band, expressing the change in percentage.

The statistical significance of the power modulation averaged across the 5 training days was assessed separately in several frequency bands of interest (namely theta: 4–7 Hz, alpha: 8–13 Hz, beta: 14–26 Hz, low-gamma: 27–40 Hz and high-gamma 41–70 Hz) at the level of scalp topography by considering the average over the 5 s of real-time BCI-control. To do so, we used a Monte Carlo test with 1000 permutations, shuffling baseline and BCI-control labels, and a 2-tailed paired t-test to compare the average power over the two intervals. We used a standard cluster-based correction to account for multiple comparisons over the scalp[50]. To assess the statistical significance of the identified clusters, their size was compared to the distribution of cluster sizes expected under the null hypothesis. Clusters that had a *p* value < 0.05 (two-tailed) were considered significant.

## Changes in EEG power across training

To assess the evolution of the neural activity during BCI-control over the 5 training days, we first averaged the power over the 5 s of real-time BCI-control, separately for each channel, day, and participant. Next, a "*LMM with planned contrast*" was fitted for each channel and frequency band separately. The statistical significance was assessed with 1000 permutations, randomly permuting which day data belonged to, within participants.

Multiple comparison correction was performed based on the same clustering method as mentioned in the previous section.

### Evolution of Brain-Behavior over 5 days of training

Next, we addressed how the relationship between the power modulation in each frequency band and BCI-control performance changes over the training period. We performed separately for each frequency band and channel, a linear mixed model with BCI performance as a dependent variable, the "*LMM with planned contrast*" and EEG power as independent variables, and participants as a random factor, leading to the following model: *BCI_performance ~ EEG_power\*day + (1|Participant)*. Significance was calculated using permutation tests followed by cluster-based multiple comparison correction (see previous section). In particular, we considered the coefficients for the interaction term "*EEG_power\*day*": a positive coefficient indicates that the effect of power on BCI-control performance becomes stronger with training, and vice versa.

### Electromyography

We assessed the potential contribution of muscular activity by computing the classifier's features and CV accuracy considering exclusively data recorded from the two EMG electrodes, separately for the *offline* and *online* session. We then compared these CV accuracies, averaged across training days, with those obtained with the EEG data using a two-tailed t-test. We tested for a linear improvement in CV accuracy computed based on the EMG data throughout training during the *online* and *offline* sessions ("*LMM with planned contrast*").

Next, we investigated differences in the evolution of the CV accuracy across the 5 days by computing the training slopes for the EMG data, and both sessions. We compared these values with those previously obtained from the EEG data (as displayed in Fig. 1e) by performing a linear mixed model analysis using the *Data* (EMG/EEG) and the *Session* (offline/online) as factors.

We assessed the relationship between the improvement in BCI-control performance and EMG activity by calculating the Pearson correlation coefficient between the slope of the CV accuracy on the EMG-*online* data and the learning slope of BCI-control performances.

Last, we evaluated the similarity between the feature's weight frequency-wise extracted with the EMG-classifier and EEG-classifier: an overlap between the frequency profiles would likely indicate strong muscular contamination in the discrimination between the two imagined syllables.

### Mental chronometry data analysis

First, we considered the average time required to complete the mental chronometry imagery and speaking task for the two modalities and ran a linear mixed model with fixed factors: the *Modality* (imagine/speak), the *Group* (continuous/discontinuous feedback), and the "*LMM with planned contrast*". This latter linear trend was further probed separately in the two groups of participants by considering the average across both modalities ("*LMM with planned contrast*").

Next, we computed an *index of deviation from isochrony* as the ratio between the average duration of the imagined and spoken syllable repetition according to the following formula already used in a previous BCI study[25] (Eq. 2):

$$deviation\ from\ isochrony = abs\left(1 - \frac{Imagine}{Speak}\right) \qquad (2)$$

In this formula, the isochrony corresponds to equal time required to perform both tasks, hence the ratio between the two modalities equals 1. According to previous studies, higher values of the *deviation from isochrony* index are expected to be associated with lower imagery skills.

We probed the evolution of this index across training and between the two groups by performing a linear mixed model with fixed effect the *Group*

(continuous/discontinuous feedback) and the "*LMM planned contrast*". The linear trend across days was further tested separately in each group.

Last, we computed the slope of the *deviation from isochrony* index by fitting, separately for each participant, a linear model considering the *deviation from isochrony* index measured on each day of training. To investigate whether BCI-learning dynamics could be reflected by isochrony, we computed the Pearson correlation coefficient between this slope and the learning slope modeling the improvement in BCI-control performance previously obtained. In addition, we calculated the Pearson correlation coefficient between the average BCI-control performance and the average deviation from isochrony to probe a relationship independent from training.

### Statistics and reproducibility

All statistical analyses were carried out using MATLAB (version R2018b, The MathWorks, Natick, MA, USA) and R version 4.1 (R Core Team, 2021). The sample size was 15 for the main experiment and 10 for the control experiment. Changes across the five days of training were analysed using linear mixed models (LMMs) with a planned contrast to model a linear trend throughout training ("*LMM with planned contrast*") as the fixed effect and participants as a random factor. At the individual level, learning was assessed by calculating the slope of a linear fit (learning slope and training slope). Differences between two conditions were evaluated using two-tailed t-tests: paired t-tests for within-group comparisons and unpaired t-tests for between-group comparisons (e.g., comparing data from the main experiment and the control experiment). Differences in performance between blocks in the main experiment were tested using one-way repeated-measures ANOVA. Relationships between two variables were assessed using Pearson correlation coefficients. The significance threshold for all statistical tests was set at 0.05.

Statistics on EEG data were performed using Monte Carlo tests with 1000 permutations, combined with two-tailed paired t-tests and cluster-based corrections to account for multiple comparisons across the scalp. The statistical significance of clusters was determined by comparing their sizes to the distribution of cluster sizes expected under the null hypothesis. Clusters with a *p* value < 0.05 (two-tailed) were considered significant. Neural changes across training were investigated using the same "*LMM with planned contrast*" approach described above.

### Reporting summary

Further information on research design is available in the Nature Portfolio Reporting Summary linked to this article.

## Results

### Training improves BCI-control abilities and decoding accuracy

Testing for a change in BCI-control performance throughout 5 training days with the continuous feedback, we observed a linear increase in average performance from Day-1 to -5 ($F_{1,59} = 5.92$, $p = 0.018$, $\eta^2_\text{p} = 0.09$, Fig. 1b), indicating that imagined speech abilities improved with training, with however marked inter-individual differences (improvement in 11/15 participants, Fig. 1c). Interestingly, we found a positive correlation between the average BCI-control performance obtained considering the entire training period, and individual learning slopes ($r = 0.55$, $p = 0.034$, Fig. 1d), showing that the best performers were those who also benefited the most from training.

As expected, there was no difference in performance between the two syllables ($T_{14} = 1.64$, $p = 0.12$, $d = 0.42$). We found a trend towards significance when testing for differences in performance between the four blocks ($F_{3,42} = 2.54$, $p = 0.07$, $\eta^2_\text{p} = 0.15$) due to a marginally lower performance during the second block than in the third ($T_{14} = 2.2$, $p = 0.045$, $d = 0.57$) and fourth ($T_{14} = 2.2$, $p = 0.043$, $d = 0.57$).

The linear performance improvement trend was confirmed by an increase in the cross-validation accuracy obtained when computing the classifier parameters on *offline* data ($F_{1,59} = 9.35$, $p = 0.0033$, $\eta^2_\text{p} = 0.14$, Fig. 1e-left, blue line). This measure exclusively considers neural data, as no

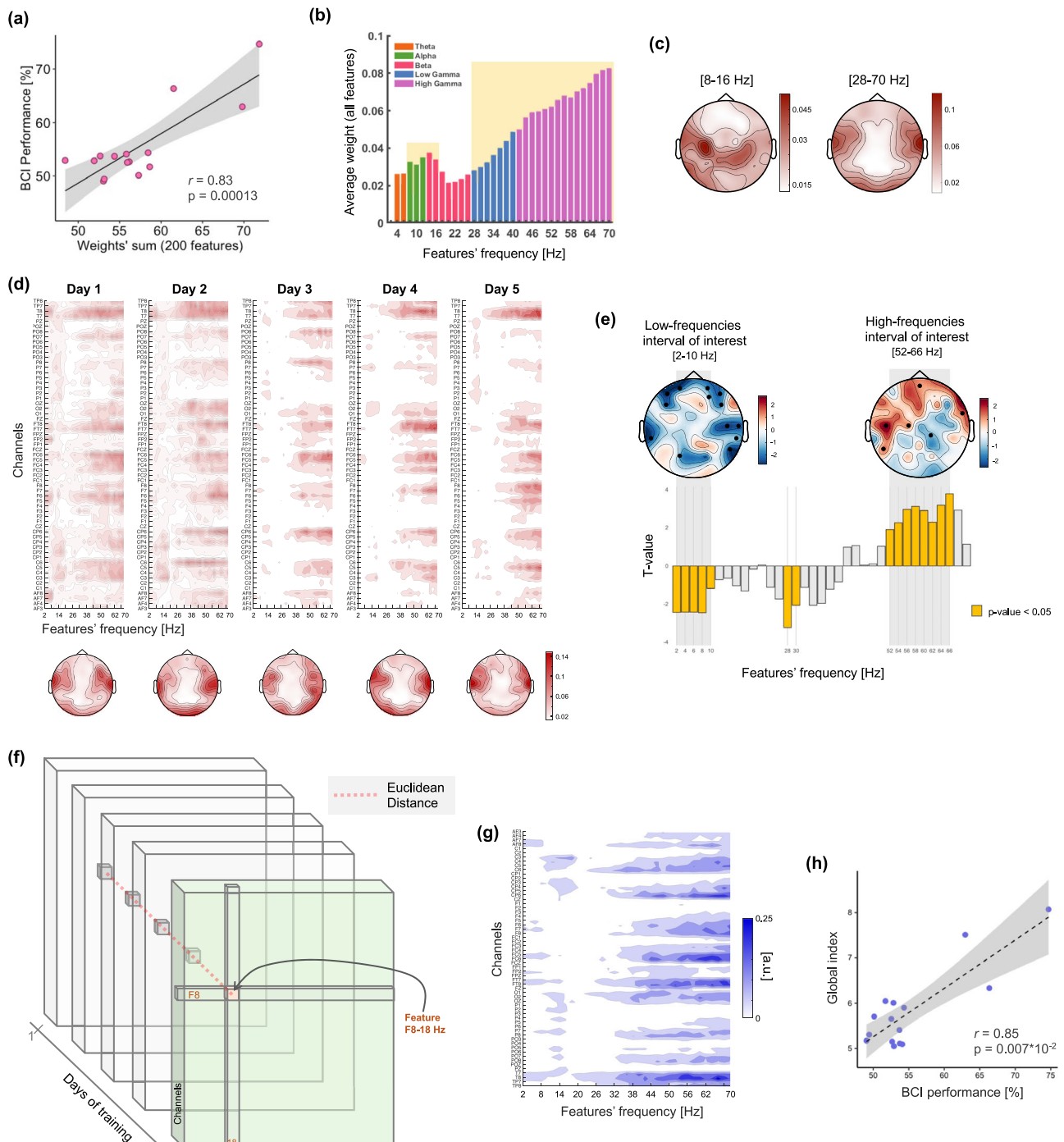

**Fig. 2 | Random forest classifier features and their evolution over training.**
**a** Relationship between individual BCI-control performance and feature weights, obtained by considering the sum of the first 200 features, averaged across the 5 training days ($n = 15$). **b** Average of features' weights for each frequency across all experimental days, with colors indicating the different frequency bands. The highlighted intervals indicate two frequency ranges with a prominent contribution to syllable decoding: 8–16 Hz and 28–70 Hz. **c** Corresponding topographies of the average weights for these two frequency intervals. **d** Maps visualizing the average feature weights across participants, for each training day. On each map, channels are represented on the y-axis, and individual frequency values on the x-axis. Scalp topographies below each map display the average weight for each channel, across all frequencies and participants, on each training day. **e** The bar plot represents statistical results (t values) when testing for changes in frequency contribution with training. Positive and negative values indicate respectively a decrease and increase in contribution throughout the training period. Yellow bars highlight the frequency(ies) for which the test was statistically significant (p value < 0.05, permutation test, $n = 15$). Scalp topographies show the results of the same statistical analysis performed on the scalp space, separately for two frequency intervals. These were defined based on the analysis in the frequency space, and include respectively the 2–10 Hz (low-frequency interval of interest) and 52–66 Hz (high-frequency interval of interest) range. A decrease in the contribution of a given electrode in the classification is indicated in blue, an increase in red (t values). Black dots highlight electrodes showing a statistically significant effect (p value < 0.05, permutation test, $n = 15$). **f** Schematics of the approach used to extract a global index to quantify the change across the 5 days of training. For each feature in the channel x frequency space, the Euclidean distance between two consecutive training days is calculated (as indicated by the red dotted line), and then the index is obtained by averaging the resulting 4 distances. **g** Euclidean distance index for each individual feature. **h** Correlation between the global index representing the amount of change both in the frequency and spatial domains (computed as the Euclidean distance between the features' weights from two consecutive days) and the average BCI performance across the 5 training days ($n = 15$).

feedback is provided during the first part of the experiment. Using the same post-processing approach, we also found a significant improvement over the 5-days of training in the *online* data ($F_{1,59} = 17.79$, $p = 8.6 \times 10^{-5}$, $\eta^2_{\mathrm{p}} = 0.23$, Fig. 1e-left, red line). Across training days, the CV accuracy was significantly higher in the *online* than the *offline* session ($T_{14} = 8.3$, $p = 8.8 \times 10^{-7}$, $d = 2.14$). This difference was not due to a difference in the learning dynamics, as we found no statistical difference between the learning slope in the *offline* and *online* sessions ($T_{14} = 1.5$, $p = 0.15$, $d = 0.38$, Fig. 1e-right). All participants except two received at least one monetary bonus, reflecting performance improvement from one day to the next.

## Analysis of the classifier's features

In a second step, we jointly analyzed participants' performance and the features used by the classifier to distinguish the imagery of the two syllables. We found a marked correlation between the individual BCI-control performance and the features' weight ($r = 0.83$, $p = 0.00013$, Fig. 2a). This indicates, as expected, that participants performing better present more discriminant features.

We then investigated which frequency bands contribute most prominently to the syllable discrimination and found a peak in contribution straddling the alpha and the lowest end of the beta interval (8–16 Hz), as well as the gamma band (Fig. 2b). Notably there was a linear increase in the average feature weights throughout the entire gamma interval, with the highest values for the highest frequencies up to 70 Hz. The topographical representation of the features according to their frequency shows that the first peak was associated with a cluster over the left central region (Fig. 2c-left), whereas the gamma band contribution originated from temporal regions, bilaterally, and posterior-occipital areas (Fig. 2c-right and Supplementary Fig. 3). A third, distinctive spatial pattern was found in the theta band, characterized by a strong contribution from frontal and temporal regions (Supplementary Fig. 3).

Importantly, the decoded frequencies associated with BCI-learning with continuous (Fig. 2b) and discontinuous feedback (Supplementary Fig. 1b) were similar, with peaks over the alpha and low-beta bands and an increasing contribution in the high-gamma band, and also showed similar topographies (Fig. 2c and Supplementary Fig. 1b). Similarities between two datasets including different participants show that attempted BCI-control based on syllables imagery engaged consistent neural features.

## Dynamics of neural features associated with BCI-control learning

First, we considered the global evolution of the features (sum of the first 200 features) and found a significant linear increase in the weights over the course of the training ($F_{1,59} = 8.62$, $p = 0.0047$, $\eta^2_{\mathrm{p}} = 0.13$, Supplementary Fig. 2b).

Next, we qualitatively inspected the change in feature weights both in frequency and spatially across the scalp. We found that across participants the most discriminant features consistently localized over temporal regions and involved most prominently frequencies above 30 Hz on each training day (Fig. 2d). While the feature weight distribution on the first two training days was rather scattered over the whole frequency-channel space, with more training it narrowed down to the most discriminant feature clusters (Fig. 2d).

To quantify this change, we first investigated the weights' evolution frequency-wise and found two frequency ranges that presented a linear change throughout the training period: while the contribution of the 2–10 Hz interval decreased, that of the 52-66 Hz interval increased with training (Fig. 2e).

To identify which regions underpinned this effect, we considered the average over each of these two intervals and ran the same analysis at the individual electrode level. We found a decrease in low-frequency contribution over bilateral temporal and frontal regions, together with an increase in the high-gamma band over left fronto-temporal regions (Fig. 2e).

We then investigated the link between the change in BCI-control performance and the change in the feature space. To do so, we extracted,

separately for each participant, a *global index* representing the amount of change both in the frequency and spatial domains, computed as the Euclidean distance between the weight of two consecutive days (Fig. 2f). By visually inspecting the Euclidean distance for each feature (averaged across participants), we observed a strong overlap of this index (Fig. 2g) and the frequency-channel feature maps (Fig. 2d). We found a strong correlation between the average BCI performance the *global index* ($r = 0.85$, $p = 0.007 \times 10^{-2}$, Fig. 2h), indicating that participants performing better during the real-time control of the visual feedback were also those whose features changed most during training.

## Power modulation during BCI-control and neural changes over training days

We subsequently investigated neural changes occurring throughout the BCI-control training considering the power modulation in each frequency band during the 5 s of BCI-control (*online* session). We first inspected changes during real-time control with respect to the baseline activity (i.e., during the last second of the fixation-cross presentation) and found a significant power decrease over frontal and left-central electrodes in the alpha band, and a similar but more widespread pattern in the beta range (Supplementary Fig. 4) together with enhanced power over posterior-occipital electrodes in the high-gamma band (Supplementary Fig. 4). Next, we investigated linear changes in power modulation throughout training. Overall power increased from day 1 to 5 on all frequency bands (Fig. 3a). In particular, theta and low-gamma bands showed the strongest and most widespread increase across training (Fig. 3a-b). Smaller clusters of power increase were also found in other frequency bands, namely alpha, beta, and high-gamma bands.

## BCI-control performance and neural changes over training

We then investigated the link between BCI-control performance and power modulation over the 5 training days. We used a linear mixed model with BCI-performance as a dependent variable and, as predictors, the power in each frequency band and the planned contrast modeling a linear trend throughout training. The analysis revealed several clusters of electrodes showing a positive interaction between the two predictors indicating that power variations in these clusters increase their impact on BCI performance with training (Fig. 3c). In other words, specific regions and frequencies show dynamic changes in the direction of a stronger contribution in determining BCI-control. These included clusters located over frontal and central regions in the theta band and over the left temporal region in the gamma band. We found additional smaller significant clusters over the central region in the alpha band, and the left posterior regions in the beta band.

## Role of real-time feedback in learning

To assess the importance of accurate real-time feedback in BCI-control improvement, we analyzed the dataset acquired with the discontinuous feedback. Unlike in the main experiment carried out using the continuous feedback, there was no significant increase in BCI-control performance throughout training ($F_{1,39} = 0.83$, $p = 0.36$, $\eta^2_{\mathrm{p}} = 0.02$, Fig. 1f).

We computed differences in *online* vs *offline* CV accuracy in the discontinuous feedback group and analyzed it together with the CV accuracy in the continuous feedback group. We found a marked linear increase in accuracy across training (main effect of *Planned contrast*: $F_{1,23.3} = 10.29$, $p = 0.003$, $\eta^2_{\mathrm{p}} = 0.31$, Supplementary Fig. 1c), higher accuracy during the *online* than *offline* session (main effect of *Session*: $F_{1,28.8} = 42.76$, $p = 3.7 \times 10^{-07}$, $\eta^2_{\mathrm{p}} = 0.6$) and a significant interaction between *Session* and *Group* ($F_{1,28.8} = 9.51$, $p = 0.004$, $\eta^2_{\mathrm{p}} = 0.6$, Supplementary Fig. 1d). We further developed this interaction with posthoc t-test and found that the marked difference in the *offline* vs *online* session was mainly driven by the higher CV accuracy in the *online* session with the continuous feedback (continuous-*online* > continuous-*offline*: $T_{14} = 8.3$, $p = 8.8 \times 10^{-07}$, $d = 2.14$; continuous-*online* > discontinuous-*offline*: $T_{23} = 2.8$, $p = 0.009$, d = 1.15; continuous-*online* > discontinuous-*online*: $T_{23} = 1.82$, $p = 0.082$, $d = 0.74$; continuous-*offline* < discontinuous-*online*: $T_{23} = 1.9$, $p = 0.067$, $d = 0.78$,

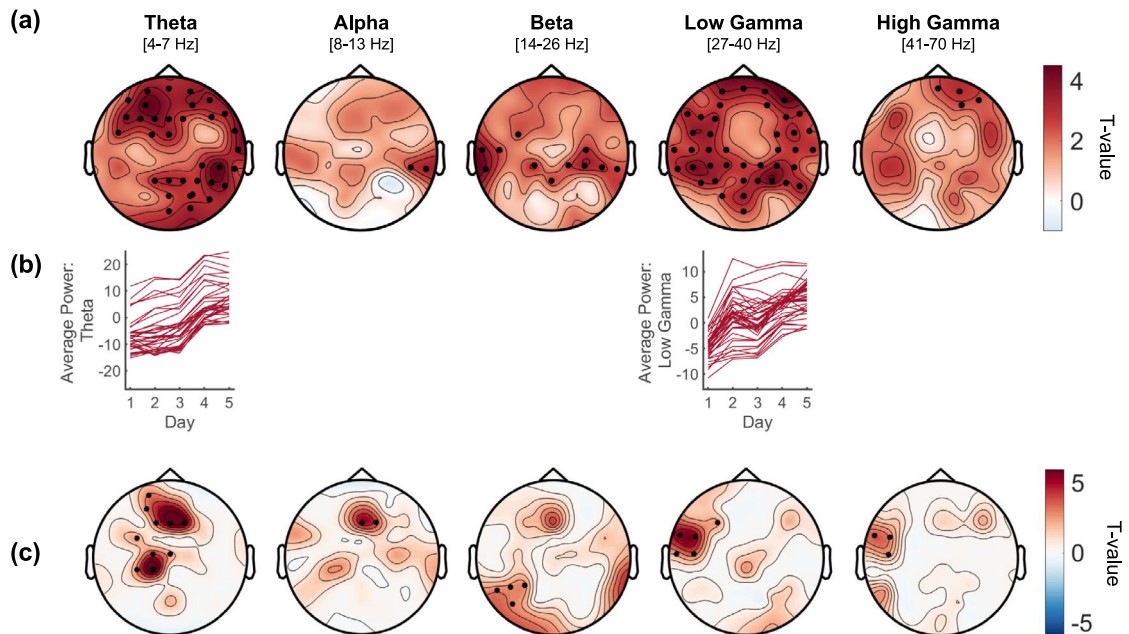

**Fig. 3 | Evolution of power modulation over the 5 training days and relation with BCI-performance dynamics. a** Topographies, one for each frequency band of interest, display the changes in EEG power across the 5 days of training. All frequency bands show a global linear power increase (positive t values). Electrodes showing significant linear trends are highlighted with black dots (permutation test, cluster corrected, $p < 0.05$, $n = 15$). **b** Power (averaged over participants) for each day, plotted for each significant channel, shows a clear increase over the 5 training days. **c** Topographies display the statistical result of the interaction term between EEG power and the effect of training (modeled with a planned contrast) on BCI-performance. Several statistically significant clusters display positive t-values, indicating that EEG power in specific regions and frequency bands became stronger with training. Among these, the most prominent were found over frontal and central regions in the theta band and the left temporal area in the low-gamma band. Significant electrodes are highlighted with black dots (permutation test, cluster corrected, $p < 0.05$, $n = 15$).

discontinuous-*offline* < discontinuous-*online*: $T_9 = 2.3$, $p = 0.04$, $d = 0.73$; continuous-*offline* < discontinuous-*offline*: $T_{23} = 0.53$, $p = 0.6$, $d = 0.21$).

## Comparison of syllable decoding from EEG and EMG

As we asked participants to imagine saying the syllables, a residual muscular activity cannot be excluded. We thus probed the potential contribution of EMG activity in discriminating between the two syllables by computing the CV accuracy of a model using exclusively EMG signals during the *offline* and *online* sessions. The average CV accuracy was above chance for both sessions (Fig. 4a). We then compared the EMG-based CV accuracy values to the EEG-based CV accuracy and found opposite effects for the two sessions. While CV accuracy for EMG data was higher than for EEG during the *offline* session ($T_{14} = 2.2$, $p = 0.044$, $d = 0.57$, Fig. 4a-left), the opposite was found for the *online* session with higher CV for EEG than EMG data ($T_{14} = 2.77$, $p = 0.014$, $d = 0.71$, Fig. 4a-right). Overall, CV accuracy based on EEG-*online* data was the highest (as compared to both EMG sessions and EEG-*offline*). Importantly, while EEG-*online* accuracy showed a strong linear increase throughout training ($F_{1,59} = 17.79$, $p = 8.6 \times 10^{-5}$, $\eta^2_p = 0.23$, as already reported at the beginning of the Results section), the same analysis performed with EMG-*online* data revealed no statistically significant change ($F_{1,59} = 2.53$, $p = 0.11$, $\eta^2_p = 0.04$). Data from the EMG-*offline* session showed that the CV accuracy increased linearly across training ($F_{1,59} = 8.69$, $p = 0.0045$, $\eta^2_p = 0.13$).

We further investigated differences in the evolution of the CV accuracy across training by computing the training slopes for the EMG data, in both sessions (Fig. 4b), as previously done for the EEG data (Fig. 1e). We found that overall, the slope of the CV accuracy was higher with the EEG data ($F_{1,14} = 4.5$, $p = 0.05$, $\eta^2_p = 0.24$), and that the interaction between *Data* and *Session* was nearly significant ($F_{1,14} = 3.36$, $p = 0.08$, $\eta^2_p = 0.19$). This latter effect was mainly driven by higher values of the learning slope in the *EEG-online* than EMG-*online* ($T_{14} = 2.45$, $p = 0.02$, $d = 0.63$), and EMG-*offline* data ($T_{14} = 1.78$, $p = 0.09$, $d = 0.46$). There was no main effect of *Session* ($F_{1,14} = 0.57$, $p = 0.46$, $\eta^2_p = 0.039$).

Next, we tested for a relationship between the slope obtained considering the EMG-*online* dataset and the behavioral improvement in BCI-control (learning slope) and found a close to significant positive correlation ($r = 0.5$, $p = 0.058$, Fig. 4c).

Additionally, we found the distribution of the average weights obtained from the EMG classifier (Fig. 4d) to be markedly different from the histogram obtained considering the EEG features (Fig. 2b). Of note, the different magnitude in the range of the average weights between the EEG and EMG feature space in Figs. 2b, 4d is due to the lower number of EMG features (there were only two EMG channels versus 61 EEG channels).

## Mental chronometry results

First, we explored differences and changes across training ("*LMM with planned contrast*") in the time required to perform the mental chronometry task considering the two modalities (imagery/speak) and the two groups (continuous/discontinuous feedback, Fig. 5a) with a linear mixed model. We found a marked difference between the two groups in the evolution over time of the task duration for both modalities (main effect of *Group*: $F_{1,23} = 10.25$, $p = 0.004$, $\eta^2_p = 0.31$, interaction *Planned Contrast × Group*: $F_{1,23} = 9.11$, $p = 0.006$, $\eta^2_p = 0.28$, Fig. 5a): participants using the continuous feedback displayed a decrease in the duration of both imagined and spoken tasks, while the opposite trend was found in the group that trained with a discontinuous feedback. There was no statistically significant difference between the two modalities and no other interactions. We further explored the linear trend separately in both groups considering the average duration across the two modalities and found a statistically significant linear increase in the continuous feedback group ($F_{1,59} = 10.76$, $p = 0.001$, $\eta^2_p = 0.15$) and a linear decrease in the discontinuous feedback group ($F_{1,39} = 10.44$, $p = 0.002$, $\eta^2_p = 0.21$).

Next, we considered the *deviation from isochrony* index: we found a statistically significant linear increase across training (main effect of *Planned Contrast*: $F_{1,98} = 5.6$, $p = 0.019$, $\eta^2_p = 0.05$, Fig. 5b) and barely significant higher *deviation from isochrony* in the dataset with continuous feedback

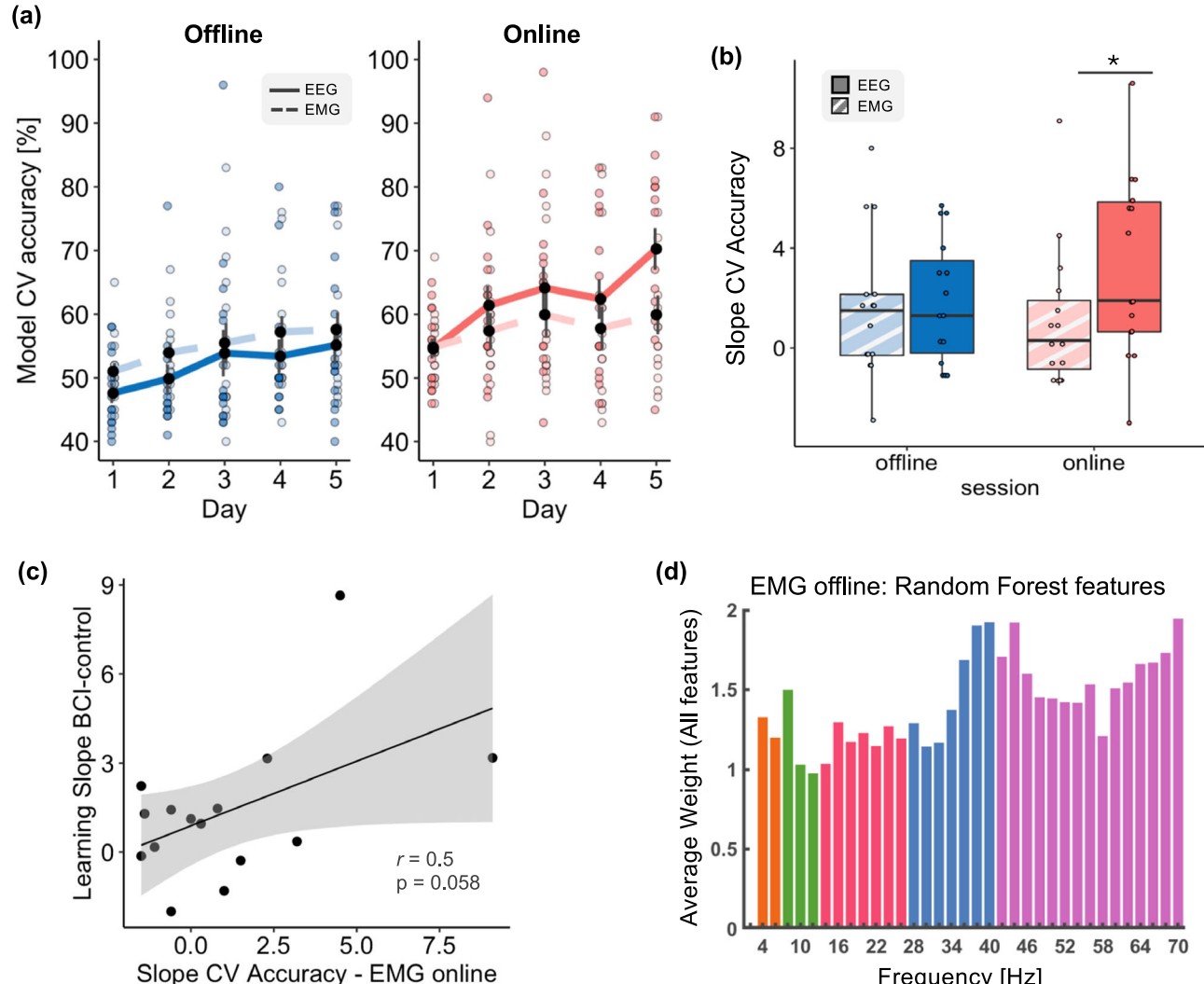

**Fig. 4 | Decoding based on electromyographic (EMG) signals.** To measure the contribution of potential muscular activity to syllable decoding, we computed a classifier based solely on EMG data from the right zygomaticus major and the orbicularis oris, separately for *offline* and *online* sessions. **a** CV accuracies based on EMG (dotted lines) and EEG (solid line) data acquired during the *offline* (blue) and *online* (salmon) sessions. Error bars indicate the standard error of the mean. **b** Training slopes calculated by fitting a linear model across the 5 training days, separately for EEG (solid filling) and EMG (striped filling) and the two sessions (*offline*: blue, *online*: salmon). Boxes represent the interquartile range (IQR), with the horizontal line indicating the median, and whiskers extending to data points that are within 1.5× the IQR from the upper and lower quartile. Individual points represent data from a single participant ($n = 15$). **c** Correlation between the learning slope representing the behavioral improvement in BCI-control and the slope obtained considering the CV-accuracy for the EMG-*online* dataset ($n = 15$). **d** Histogram representing for each frequency, the average features' weights obtained considering the EMG activity recorded during the *offline* session (the histogram is to be qualitatively compared with Fig. 2b). Significance is denoted with * for $p < 0.05$.

than in the discontinuous feedback one (main effect of *Group*: $F_{1,23} = 3.96$, $p = 0.058$, $\eta^2_P = 0.15$, Fig. 5b). The *Planned Contrast* x *Group* interaction was not significant ($F_{1,98} = 0.003$, $p = 0.95$, $\eta^2_P = 3.54 \times 10^{-05}$).

We further assessed the linear trend across training separately in the two groups and found that the linear increase was statistically significant in the group with discontinuous feedback ($F_{1,39} = 5.25$, $p = 0.02$, $\eta^2_P = 0.12$) but not in the group with continuous feedback ($F_{1,59} = 2.46$, $p = 0.12$, $\eta^2_P = 0.04$).

Next, we tested for a relationship between BCI-control and the *deviation from isochrony* and found no statistically significant correlation neither considering the learning slopes of BCI-control performance and isochrony ($r = 0.051$, $p = 0.81$, Supplementary Fig. 5a), nor between the average across the training days ($r = -0.08$, $p = 0.69$, Supplementary Fig. 5b).

## Discussion

The study shows that healthy individuals can learn to control an EEG speech-BCI by training over 5 consecutive days, and uncovers the neural

mechanisms related to the acquisition of BCI-control skills. Learning to operate a BCI based on covertly executed tasks has hitherto been investigated almost exclusively in the motor domain[52–54]. In the field of speech-BCIs, previous studies show that it is possible to improve operating an intracranial BCI through attempted speech[34] but not yet via imagined speech. Here, we found that real-time BCI-control performance increased from 55% to 70% over the 5 training days when participants received a real-time and accurate feedback, and that this increase was paralleled with higher discriminability of the neural signals (CV accuracy) during the *online* than *offline* session. These results demonstrate that closing the loop is essential to the learning process. In addition, using a discontinuous feedback, we show that accuracy and consistency of the real-time feedback are key features to achieve optimal performance. Feedback alteration consisted of visually informing the subject only when the decoded syllable was the cued one; displaying exclusively successful changes resulted in a discontinuous feedback. Better learning with continuous than discontinuous real-time feedback aligns with previous observations made during motor-BCI training[55,56]

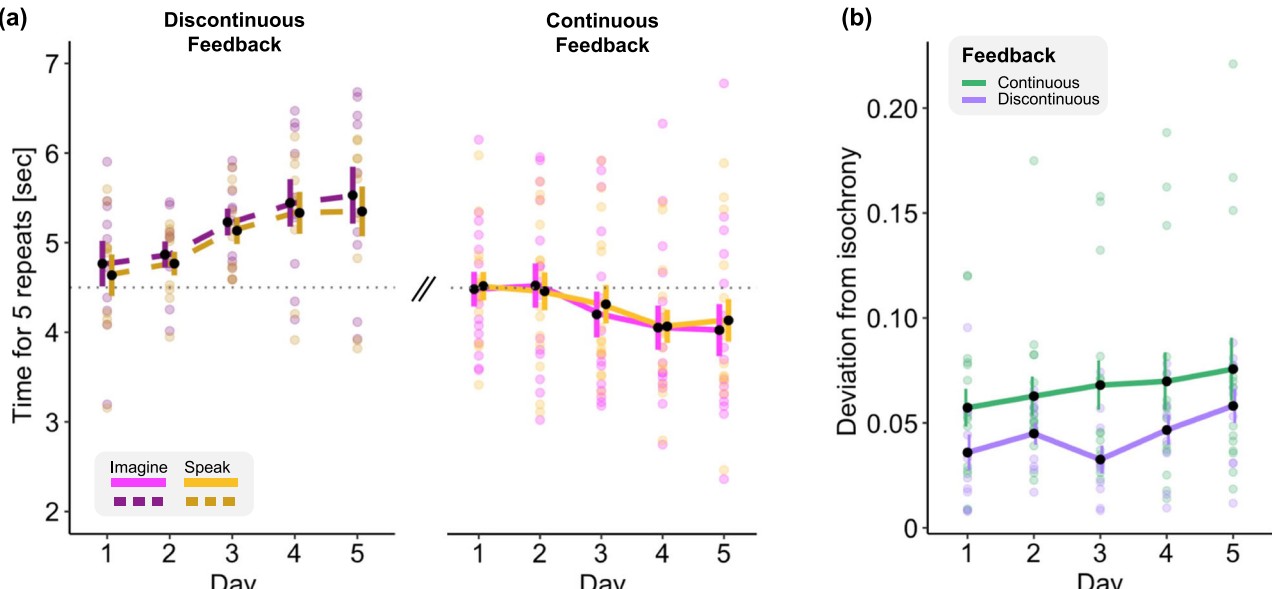

**Fig. 5 | Mental chronometry results. a** Time required to repeat 5 times one of the two syllables for the imagery (magenta) and speaking (yellow) modalities in the group with continuous (lighter colors) and discontinuous feedback (darker colors) over the 5 training days. **b** Deviation from isochrony index reflecting the ratio between the average duration of the imagined vs spoken syllable repetition, in the group that trained with continuous (green) and discontinuous (purple) feedback. Error bars indicate the standard error of the mean.

and indicates that accurate and high-rate feedback enables an optimal error-driven strategy to control the BCI.

Although the majority of the participants who trained with the continuous feedback (11 out of 15) improved their performance, there were marked inter-individual differences both in control skills and in learning slope, extending to speech-based BCI-control the phenomenon of "BCI-illiteracy", well-known in motor-imagery BCIs[24,36,57]. Poor BCI-operability seems to affect even more severely imagined speech than attempted speech, as on the first training day performance was below chance in most participants, with no outstanding performer. This effect is likely due to several factors including the relatively weak neural signals elicited by speech imagery[14,17], their limited spatial separability, and the restricted access to deeper speech brain regions with surface EEG, a situation that sharply contrasts with the easily decodable focal and superficial patterns elicited by hand motor imagery. Quite predictably, steeper learning was found in better performers. A dichotomy in learners versus non-learners has previously been reported during a single training session of volitional control of individual neurons in mnemonic structures[58]. Here, we show that this dichotomy remains present when training is carried over a longer time period but can however be mitigated by repeating the task over multiple sessions.

Individual factors likely play an important role in acquiring BCI skills, including cognitive, affective, and somatic aspects[25,59]. A critical factor in determining BCI-control is arguably a cognitive one, namely the ability to imagine syllables. We thus probed whether learning to control the BCI was accompanied by changes in imagery abilities, and attempted to quantify it using the mental chronometry task. According to this test, good imagery skills would be reflected in an equal time (i.e., isochrony) to perform the imagery and the overt execution. As suggested by a previous motor imagery study[60], isochrony is expected to be associated with improved BCI-performance over training. The isochrony hypothesis was not confirmed in the group who used the continuous feedback, however, the time taken to repeat imagined and spoken syllables decreased with training, indicating that the task became easier over the training days[61]. This facilitation effect might partly underpin the BCI-control improvement in this group. Interestingly, in the group that employed the discontinuous feedback, the difference between imagery and execution increased with training, along with the time taken to repeat the syllables.

These only partly conclusive findings and the lack of a correlation between mental chronometry and BCI-control performance suggest that the isochrony test is probably not the best index to quantify imagined speech learning, as it might be affected by a ceiling effect due to the hyper-automaticity of syllable repetition.

Given extended reports indicating residual EMG activity during inner speech and even the possibility of above-chance EMG decoding (see for a review[62]), we tested whether EMG signals alone could be classified by computing the CV accuracy in post-processing. The underlying hypothesis is that learning effects should be absent or at least less pronounced in EMG than in EEG data. Consistently, we found no decoding improvement during the *online* session with EMG, and no distinctive decoding features. However, the above-chance decoding on some of our EMG datasets and a significant increase in CV accuracy *offline*, indicate that speech imagery was likely accompanied by subthreshold motor activation, a finding that is compatible with the fact that participants were instructed to imagine pronouncing the syllables (rather than e.g., hearing syllables). The presence of residual EMG activity during mental imagery has been the subject of debate for almost a century[63,64] and there are still contrasting results in the field of speech imagery[42,62,65–67]. Our results are in line with the *Motor Simulation View* (in contrast with the *Abstraction View*) of inner speech, where peripheral muscular activity would result from imperfect inhibition of motor commands[62,64], possibly accounting for the selectivity of EMG signals to specific phonemes[42,68–71]. Above-chance decoding found on average on some days confirms that EMG activity is more than a merely non-specific tonic activation[42,66], and is subject to marked inter-individual differences[42]. EMG activity during imagery is also modulated by the intensity of the mental effort[64,72], an effect that given the participants' reported experience, likely contributes to the observed correlation with the learning slope. Critically, during the *online* session, EEG-based decoding achieved significantly higher accuracy than EMG-based decoding, and showed an improvement across training, unlike EMG-based decoding. This, along with the distinct patterns of decoding frequencies, indicates that potential contamination of EEG signals by EMG activity (either as muscle artifacts or neural activity elicited by overt speech) did not interfere with the acquisition of BCI-skills. While EMG signals likely contained information about the syllable choice, learning involved changes occurring at the level of neural activity elicited by covert speech. Significant learning effects might also be observed by providing a

feedback based exclusively on EMG, given the high-decoding accuracy achieved with a speech-BCI based on EMG signals[66]. This kind of closed-loop system could benefit patients who retain some residual orofacial movements, and for non-invasive solutions, might have some efficacy. This question remains however outside the scope of the present study.

Here, one of the main goals was to explore the evolution of neural features throughout the learning process. The BCI-control improvement was accompanied by specific changes in the decoding features and in the EEG power. On average, the most discriminant features were located over the temporal regions bilaterally in the gamma band, and over the left sensorimotor cortex in the 8–16 Hz range, overlapping with key speech areas[37,73] previously exploited as decoding sites[6,29]. These neural features were qualitatively similar whether the subject got continuous or discontinuous online feedback, indicating that the basic set of neural features mobilized by operating a BCI with syllable imagery was independent of the experimental specificities. The feedback dynamics however was key to the learning process.

Over the 5 training days, we qualitatively observed a pruning effect within the features' space, with the least discriminant features progressively decreasing their contribution in favor of more focal clusters around the features contributing most to the classification. Specifically, lower frequencies in frontal and temporal regions decreased their contribution in favor of a stronger involvement of the high-gamma band in frontal and left centro-temporal regions. Similar pruning effects have been observed with fMRI-neurofeedback training, resulting in a reduction of redundant connections while strengthening the relevant ones in a restricted set of brain regions[74]. Whether this effect is more pronounced when the action is performed without motor output, such as in an imagined speech BCI, has yet to be determined. Importantly, higher BCI-control over the 5 training days was associated with stronger changes in the feature space, indicating that features' dynamics play a crucial role in the learning process. From a technical viewpoint, this implies that the classifiers should be set to grasp the individual learning profile by a dynamic calibration of their parameters while the BCI is being operated, such as with adaptive classifiers that are able to account for learning-related changes in real-time[75].

The power of the neural activity elicited by BCI control (irrespective of the imagined syllable) also substantially increased with training over the entire spectrum, most prominently in the theta and low-gamma band. Both frequency bands are highly relevant in speech perception and production, respectively underpinning syllabic and phonemic processing[76,77]. Interestingly, we found that these same two frequency bands were prominently influencing BCI performance as learning progressed, specifically over fronto-central regions for theta power and the left temporal area for the low-gamma band. The fact that the contribution of the theta band as a decoding feature decreased (Fig. 2e) while undergoing substantial power increase across training (Fig. 3a) in relation to performance improvement (Fig. 3c) might appear contradictory. The increase in theta power might however reflect non-syllable specific mnemonic encoding[78] via changes in synaptic plasticity[79] known to occur over the same time scale as the training duration in our experiment[80]. Further analyses are necessary to elucidate a potential top-down role of the theta band on higher frequency bands involved in discriminating between the two syllables.

The present study fills an important gap in the field of imagined speech BCI, which traditionally suffers from low performance, by showing that controllability can be improved with training, even when starting from chance-level performance. It provides solid neurophysiological grounds to improve current BCI systems based on speech-imagery, notably by enhancing decoding using a pre-defined subset of brain regions over temporal and fronto-central areas and frequency bands, which we found to be implicated in both decoding and learning. Although surface EEG-based BCIs are unlikely to become stand-alone communication devices, they will find valuable applications in the field of neurotechnology for language disorders. Training to perform real-time control could be used to select those patients who would benefit most

from invasive BCIs for communication[81]. To validate such an approach, future work is required to 1- establish the correspondence between surface and intracranial EEG recordings of the neural activity elicited by imagined speech, as previously suggested for sensorimotor rhythms in patients with locked-in syndrome (LIS)[82], and 2- take into consideration the potential reduced BCI-controllability in patients as compared to healthy users[83]. Indeed, BCIs based on imagined speech might not be suitable for all patients in the long term, such as those in which motor impairments are accompanied by progressive cognitive decline (e.g., LIS[84]). They might however benefit many individuals in whom these functions are spared, for instance, patients with post-stroke aphasia, where attention and global control are generally preserved[85] and importantly, imagery skills are better retained than spoken language[86–89]. In these patients, rehabilitative interventions based on closing the loop on imagery attempts with real-time feedback could be expected to mobilize residual neural patterns and promote neural plasticity. Importantly, such interventions will have to be adapted to each individual's residual speech ability and specific impairment.

The future of speech-imagery BCIs holds promise for a variety of purposeful scenarios, particularly those that rely on human-machine co-adaptation.

## Data availability
Source data underlying the graphs can be found at this link: https://osf.io/vr26k/. Raw data that support the findings of this study are available from the corresponding author upon reasonable request.

## Code availability
Custom-made scripts for the analyses can be found at this link: https://osf.io/vr26k/ [90].

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

## Acknowledgements

We thank the Human Neuroscience Platform of the Fondation Campus Biotech Geneva and Shizhe Wu for technical advice. This study has been supported by the National Center of Competence in Research "Evolving Language", Swiss National Science Foundation Agreement #51NF40_180888 and the Fondation pour l'Audition.

## Author contributions

Kinkini Bhadra: conceptualization, methodology, software, investigation, data curation, formal analysis, writing - original draft and editing, visualization. Anne-Lise Giraud: conceptualization, supervision, writing – reviewing and editing, funding acquisition. Silvia Marchesotti: conceptualization, methodology, software, formal analysis, data curation, writing – original draft and editing, visualization, supervision.

## Competing interests

The authors declare no competing interests.
