## [Transparent Peer Review file · Communications Biology]

Learning to operate an imagined speech Brain-Computer Interface involves the spatial and frequency tuning of neural activity

Corresponding Author: Dr Silvia Marchesotti

Version 0:

Reviewer comments:

Reviewer #1

(Remarks to the Author)

Review of Bhadra et al: Learning to operate an imagined speech Brain Computer Interface (BCI)....

The ms describes an experiment with 15 healthy young persons over 5 two-hour sessions to learn to control an EEG-based BCI by using imagery of two different syllables. The decoder was adapted individually in a baseline off-line session during the imagery of the two syllables without real-time feedback. The decoder classifying the two images best was then used for the on-line feedback session presenting visual feedback on a screen. EMG contributions were controlled by two face electrodes. The authors argue that, demonstration of learning of speech imaging control with an EEG_BCI could be useful for patients suffering from aphasia and other neuronal language disorders or locked-in syndrome. They do not cite evidence that imagery may disturb or hinder operant BCI-control in non-invasive and invasive BCI-control in patients (see the work of Seguin and Mattout, several papers over the last years)

A clear learning of BCI control in about 10 of the 15 subjects particularly in the theta and low gamma range was found. The separation between the two images also improved. The authors interpret their finding as evidence that BCI-control improves with learning of linguistic imagery and vice versa probably. Since they did not employ a patient sample the generalisation of the results to pathological conditions remain unclear. Non-invasive BCIs using EEG or NIRS or fMRI work well with healthy people but in patients they often do not work as can be seen in the large BCI-literature (see several reviews by the above mentioned authors Seguin, Mattout or others i.e. in Nature Rev.Neurology or Sitaram in Nature Rev Neurosc.). Thus, demonstration in healthy persons is of limited clinical value. Any other type of imagery, particularly involving motor elements, whether linguistic or not have similar learning trajectories and no imagery at all may be optimal relying on the feedback alone. A control condition with no imagery instruction and no instruction, just asking for attention to the positive feedback usually works best (see also animal work of Koralek et al in Nature) but was not used here.

Reviewer #2

(Remarks to the Author)

1. What are the major claims of the paper?

A binary imagined speech BCI with EEG can improve with training across multiple days, though not for all participants, similar to the larger (hand) motor imagery literature. The improvement is linear. Theta (frontal and central) and gamma (left temporal) features are most related to the improvement in performance.

2. Are the claims novel? If not, please identify the major papers that compromise novelty. Will the paper be of interest to others in the field? Will the paper influence thinking in the field?

Yes, I think this paper is important as it highlights the power of training especially on the user-level. The study includes a lot of data, which is very valuable.

3. Are the claims convincing? If not, what further evidence is needed?

The claims are pretty convincing, though additional information would make it stronger.

The task section 'syllable imagery' should include the fact that participants were asked to do the imagery for 5 seconds continuously (?), and whether or not that meant repeating imagining saying the syllable many times.

Along the way, participants may have changed the strategy they used to control the BCI (imagine the syllable) towards a typical neurofeedback strategy, or anything that seemed to work to modulate their brain rhythms, for example. I would suggest the authors to include a report of what strategy the participants used during the experiment, retrospectively.

The EEG-EMG analysis did not include a test on the linear increase throughout training on the offline EMG data. This does seem to be significant so I would suggest the authors to report it along with some discussion on what this could indicate. For both offline and online analyses, they (EEG and EMG) do seem to follow the same trend. I also wonder if there was a difference between the responders and non-responders (did the responders have more motor activity perhaps?).

4. Are there other experiments that would strengthen the paper further? How much would they improve it, and how difficult are they likely to be?

A comparison experiment where the task instruction is modified could be an interesting control experiment. However, I don't think that is necessary for the current study to be valid.

5. Are the claims appropriately discussed in the context of previous literature?

Yes.

6. Is the manuscript clearly written? If not, how could it be made more accessible?

The manuscript is clearly written for the most part. The introduction and methods could use a grammatical review (with a focus on the use of prepositions and plurality, specifics are in the annotated file). The second sentence of the abstract is very long, so I would suggest to split this up into separate sentences. The methods section was a bit difficult to read and understand on the first iteration. I would suggest to make it more concise, avoid repetition and be more clear in defining terms.

7. Could the manuscript be shortened to aid communication of the most important findings?

See previous comment, the methods may be shortened.

8. Have the authors done themselves justice without overselling their claims?

Yes, although the abstract can be toned down a bit. In line 26, it was stated that the findings indicate that learning 'must' be considered. It definitely shows it's a very important factor, but I do not see that the paper provides evidence that it's a 'must'. In the same sentence, it was stated that the findings indicate "that non-invasive BCI-learning can help predict the individual benefit from an invasive speech BCI". This statement would require a study on both EEG and invasive measurements, so I would not phrase this as an outcome of the current study, but rather a point for further research (for example, see Mansoureh Fahimi Hnzaee et al., 2022, J. Neural En; DOI: 10.1088/1741-2552/ac8764). Similar to how it was written in the actual conclusion section.

9. Have they been fair in their treatment of previous literature?

Yes.

10. Have they provided sufficient methodological detail that the experiments could be reproduced?

Yes.

11. Is the statistical analysis of the data sound?

As far as I can tell the statistical analysis is sound.

12. Should the authors be asked to provide further data or methodological information to help others replicate their work? (Such data might include source code for modelling studies, detailed protocols or mathematical derivations).

I would always suggest to provide a link to the scripts used for the analysis.

13. Are there any special ethical concerns arising from the use of animals or human subjects?

No, the human subjects appear to have been well informed.

14. Additional comments?

Please note the annotated manuscript for more detailed comments.

Reviewer #3

(Remarks to the Author)

The present study is devoted to further understanding how imagined speech could be decoded from surface EEG recordings using a BCI system that could translate these recorded signals into text or synthesized speech. This is an exciting field and many new advances have been produced in recent years. More in particular, decoding imagined speech could be important for helping patients in which speech production areas have been damaged (e.g., post-stroke aphasia; recent research have shown preserved inner speech in some of these patients that could benefit for this type of decoding techniques). In order to investigate more on this issue, the authors created a close-loop BCI system (relying on scalp EEG signals) that was proved to decode two imaged syllables (with clear phonetic dissimilarities that could maximize decoding capacity), in 15 participants, trained during 5 consecutive days. The experiment consisted in two phases, an offline experiment, where the authors collected data to train the classifier; and an online experiment, where they use the fitted classifier to predict the imagined speech. Their results show a slightly above chance accuracy considering the large interindividual variability, and certain amount of learning. Overall, although this is an important and well conducted experiment but many concerns arise when carefully reading this article.

Some major concerns about the study:

1. The main concern is related to the task used for the experiment. The task created in this study for imaged speech consisted in repeating an instructed syllable during 5 seconds, and this information is further used for training the BCI and further decoding. However, it is unclear how much repetitions are produced during this period internally, and it seems a very uncontrolled experimental situation (for example, variability might exist on how long where the syllables produced by each participant, where the participant following a particular rhythm, how many syllables produced each participant, etc.). As the imaged internal speech is not properly structured, it is difficult to correctly train the classifier (indeed, part of the EEG epoch might not correspond to the expected internal speech adding noise to the signal). For example, a better approach would have been to repeat only once each syllable (e.g., 1 s internal slow production), and therefore, ensuring that the whole EEG epoch would correspond to the internal generated signal (reducing noise of silence or non-production stages). As it is right now it is difficult to see how this uncontrolled setting could derive in a good classification pattern and further learning across days. A recent study (see Mor Regev and colleagues, in *Cer. Cortex*, "Mapping Specific Mental Content during Musical Imagery") tried a very interesting approach to study for first time music covert productions. They pre-trained participants with a particular melody (particular rhythm) and later in the scanner, the requested participants to mentally replay this melody internally. In this situation, the correlation analysis between the real signal and the internal production (fMRI analysis) allowed them to infer the quality of the signal reproduced internally. Similar settings could be designed to ensure a larger degree of control about the overt or internal speech signal. To my view, the design of the task was not sufficiently innovative and the authors might need to think more deeply on these important issues.

2. Authors used a random forest procedure to decode the syllable imagery. Random forests could be a good option for the offline part, since they are fast to train, but there are slow to test (the multiple decision trees have to predict the label of the test data). Therefore, these classifiers are not the best ones to perform real-time predictions. In fact, authors did not report the delay between data collection, in the online experiment, and the given feedback.

3. Authors considered the weights of the first 200 features, based on the cumulative sum, but looking at Suppl. Fig 1, the elbow (around 75% of the cumulative sum) is found around 400 features.

4. As authors stated, there is an increase of the CV accuracy across sessions, but the accuracy is surprisingly low. Indeed, it is around chance level (chance = 50%) in almost all offline sessions (blue line in Fig. 1E). As the authors discussed, the CV accuracy improved in the online session, but this accuracy is the result of an offline classification of the online data. With these results, it is not surprising the low performance of the BCI system (Fig. 1C). In fact, something that I wonder is whether the participants were upset with the system because it continuously failed. Did the authors asked participants for feedback? This is another aspect that need to be improved, asking participants to provide a report of their internal production as well as opinions about the experiment in general.

5. Another surprising result is the EMG accuracy (Fig. 4), which is very similar to EEG accuracy in the offline experiment, and it is slightly lower in the online experiment. The EMG pipeline considered just 70 features, while EEG pipeline considered 2,135 (30 times more features than EMG). I think that these results should be more detailed in the discussion section.

Some minor concerns about the study:

1. There is a lack of machine learning terminology. For instance, the authors stated "classifier calibration", while classifiers are usually "fitted" or "trained".

2. The battery in the offline experiment is increasingly fulfilled, but it is not specified how this battery is fulfilled in the online experiment (I guess it is binary: fulfilled or empty).

3. In the test for the linear increase in CV accuracy (using LMMs), authors could consider the interaction between planned contrast and sessions, instead of building separated models up.

4. A detailed rationale of the global index is necessary.

5. Report the exact p values of statistics.

6. Highlight the chance level in Figure 1C.

7. Make labels in Fig. 2 larger, please.

8. Write the full references in the reference section, please.

Version 1:

Reviewer comments:

Reviewer #1

(Remarks to the Author)

The authors changed their manuscript according to the reviewer's criticism and included data of a control experiment

demonstrating the improvement in performance of the imagery BCI.
They should check their citations again, there are several (minor) errors:
Metzger 2023 lacks volume info
Saroush et al wrong Vol
Maiseli et al lacks page info
Panachakel et al Journal missing
Panachakel 20221 all info missing
Nguyen et al lacks pages
Steyrl et al spelling mistakes
Ostenveld et al pages

Reviewer #2

(Remarks to the Author)

The authors made major revisions to the manuscript and added additional control experiments, for which I commend them. Unfortunately, the manuscript is still rather convoluted and lacking in logic, therefore not yet on par with the quality that I would have expected.

Following are a few more specific comments with regard to the rebuttal of my previous comments.

1. I appreciate the mental chronometry inclusion and additional information regarding the mental strategy participants used. The authors mention in the first comment that participants imagined the syllables 5 times in roughly a 5-second period for the mental chronometry. Was it also the instruction during the syllable imagery, to repeat it 5 times, approximately 1 syllable per second? This information should be added to the manuscript, for both tasks I don't see the *5* repetitions mentioned in either of the tasks. For clarity, I would also suggest to move the syllable imagery task explanation from the 'Offline session and classifier calibration.' section back to its own section underneath the experimental paradigm and mental chronometry sections.
2. In the first comment, it was also mentioned that "Interestingly, the single participant who reported a different strategy was the individual displaying the lowest (negative) learning slope.". I disagree with this statement, as this participant also reported imagined articulation and other participants reported multiple additional strategies. A better argument would be the (trending) negative correlations between performance and syllable imagery rate, perhaps taking individual data points rather than the average of 5 days can aid in the correlation.
3. I am satisfied with the additional tests and discussion on the EMG data.
4. A control experiment was added in which the "the real-time feedback was discontinuous, i.e. not systematically related to the classifier output and displaying only positive changes", the reasoning for this was explained in the supplementary material. However, there was no explanation given for the addition of an "auditory cue similar to the sound of a metronome, imposing a pace for the syllable repetition arbitrarily set at 1.4 Hz". Could this have anything to do with the odd finding in figure 5 that the first continuous feedback group got faster over time, whereas the new discontinuous feedback group slowed down over time?
5. The authors provided additional rationale for a focus on kinesthetic sensation of the current paper in the methods section. However, this doesn't take away my concern that it goes in stark contrast to what was written in the introduction: "Recent years have seen great advances in the field of speech-BCIs, most often through the decoding of motor representations of vocal tract movements from intracranial electrophysiological recordings 3–8, which have led to impressive decoding speeds reaching about 78 words per minute 4. Such an approach, however, is unlikely to be suitable for disorders of language where speech production areas are damaged, such as in post-stroke aphasia. A BCI appropriate for these disorders would require decoding representations of speech units produced through imagined, rather than attempted speech, in particular involving the language temporo-frontal system".

Additionally, the authors added a large section on deviation from isochrony index of the chronometry task of which I do not see the added benefit considering the already lengthy manuscript. Furthermore, the authors included many correlations of which they do not report the r and p values in the figures themselves and they must be interpreted very cautiously considering the very small sample size ($N=15$).

Reviewer #3

(Remarks to the Author)

The authors did a great job revising the comments I made, also the responses in general were very well developed. I do not have further comments and thanks for taking into account these concerns.

Version 2:

Reviewer comments:

Reviewer #1

(Remarks to the Author)

I checked the rebuttal letter and the manuscript and for the three reviewers; my impression is that the authors carefully responded and changed accordingly. My own review is not in the list you sent me, maybe they responded already. In general I regard this manuscript not as of high priority: the authors conceptualize their experiment as relevant for BCI-training in

patients with communication disorders. However, the article ignores completely discussing or even mentioning that a substantial amount of literature at least of locked-in patients has completely failed to generate any verbal communication with EEG-BCI except for "yes" and "no" signals and only invasive recordings allowed verbal communication (see Chaudhary et al Nature Commun. 2022 and others before). Thus the approached here presents some interesting theoretical results but seems to be irrelevant for any future clinical applications of EEG-BCI, in contrast of what they emphasize several times in the manuscript.

However, from a formal point of view, the tone of the three reviews you sent me is positive.

Reviewer #3

(Remarks to the Author)

I have reviewed the answers to Reviewer 2. I think the authors responded appropriately, always providing new information and specially clarifying the problem mentioned in point 2, regarding the instructions in the imaginary tasks, which indeed, there was not concrete instructions (this has been now clarified in the text).

Overall, considering all the revision process and current answers, the quality of the paper increased a lot, being the whole review process very constructive.

Reviewer #1

Review of Bhadra et al: Learning to operate an imagined speech Brain Computer Interface (BCI)....

The ms describes an experiment with 15 healthy young persons over 5 two-hour sessions to learn to control an EEG-based BCI by using imagery of two different syllables. The decoder was adapted individually in a baseline off-line session during the imagery of the two syllables without real-time feedback. The decoder classifying the two images best was then used for the on-line feedback session presenting visual feedback on a screen. EMG contributions were controlled by two face electrodes. The authors argue that, demonstration of learning of speech imaging control with an EEG_BCI could be useful for patients suffering from aphasia and other neuronal language disorders or locked-in syndrome. They do not cite evidence that imagery may disturb or hinder operant BCI-control in non-invasive and invasive BCI-control in patients (see the work of Seguin and Mattout, several papers over the last years)

A clear learning of BCI control in about 10 of the 15 subjects particularly in the theta and low gamma range was found. The separation between the two images also improved. The authors interpret their finding as evidence that BCI-control improves with learning of linguistic imagery and vice versa probably. Since they did not employ a patient sample the generalisation of the results to pathological conditions remain unclear. Non-invasive BCIs using EEG or NIRS or fMRI work well with healthy people but in patients they often do not work as can be seen in the large BCI-literature (see several reviews by the above mentioned authors Seguin, Mattout or others i.e. in Nature Rev. Neurology or Sitaram in Nature Rev Neurosc.). Thus, demonstration in healthy persons is of limited clinical value. Any other type of imagery, particularly involving motor elements, whether linguistic or not have similar learning trajectories and no imagery at all may be optimal relying on the feedback alone. A control condition with no imagery instruction and no instruction, just asking for attention to the positive feedback usually works best (see also animal work of Koralek et al in Nature) but was not used here.

Answer. We would like to start by thanking reviewer 1 for agreeing to review our manuscript and for bringing up the clinical validity of our study, more specifically the suitability of using an imagery task in patients affected by severe motor impairments. We have now reviewed all studies mentioned by Reviewer 1 (references below) and we can confidently answer that these claims only partially apply to our study. The interesting hypothesis developed in this article series posits that the severe motor impairments in global paralysis and locked-in syndrome (LIS) also impact cognition by altering selective attention, ultimately preventing BCI-control. While these claims apply to LIS and perhaps ALS to some extent, the clinical population targeted by a speech-BCI is broader and includes other conditions such as cerebral palsy, and disorders of language such as aphasia, which is not a pure "motor" condition. In aphasia, attentional cognitive deficits can be impaired or preserved, and the type and severity of cognitive symptoms are widely heterogeneous [Murray, 2012], differentially impacting recovery [Brownsett et al., 2014]. Importantly, imagery skills are likely better preserved than spoken language, as demonstrated by several previous studies [Fama et al., 2017; Fama and Turkeltaub, 2020; Sierpowska et al., 2020; Stark et al., 2017]. In these patients with brain damage to speech production areas (cortical strokes) and in others with e.g. cerebral palsy in which language is disrupted by important dyskinesia, invasive BCI could represent in the future the primary choice to restore communication. The effectiveness of these approaches has been established in two studies published in Nature last year, respectively in one patient affected by amyotrophic lateral sclerosis (ALS) who retained some limited orofacial movements [Willett et al., 2023], and in another patient affected by severe paralysis and anarthria following a brainstem stroke [Metzger et al., 2023]. Based on these promising results, we believe ruling out the use of imagery-based speech-BCIs as a whole would preclude many patients from benefiting from significant life quality improvements, especially given the differences in cognitive impairments [Murray, 2012].

We concur with Reviewer 1 on the importance of assessing the feasibility of the cognitive task chosen to control the BCI. The choice should be guided by the residual abilities of each specific clinical population, for instance in LIS patients, tasks relying on attention (as in one of the studies indicated by the Reviewer) and decoding from alpha and beta rhythms might not be appropriate. Previous studies show that these patients fail to exhibit alpha or beta peaks [Höhne et al., 2014] and present a weakness in alpha power modulation [Hnazaee et al., 2022] possibly associated with cognitive decline [Lejko et al., 2020]. Thus, BCIs aiming at decoding motor imagery or attention are likely to be less effective in

patients with reduced alpha activity. Rather relying on higher frequency bands such as those shown to be important in our experiment (i.e. beta and gamma-band) and avoiding decoding purely attentional neural correlates, might preserve BCI controllability.

To acknowledge that LIS is perhaps not the best indication for a BCI based on speech imagery, we have rephrased the related sections and cited the interesting work by Séguin and collaborators. We also understand the reviewer’s concern about the need to empirically generalize the findings from healthy participants to clinical populations and taking into consideration comment #8 from Reviewer 2, we have added the following text in the conclusions (Page 23, lines 778-786):

“Training to perform real-time control could be used as a benchmark to identify learners who would benefit most from intracortical BCIs. To validate such an approach, future work is required to 1- establish the correspondence between surface and intracranial EEG recordings of the neural activity elicited by imagined speech, as previously suggested for sensorimotor rhythms in patients with locked-in syndrome (LIS) [Hnazaee et al., 2022], and 2- take into consideration the potential reduced BCI-controllability in patients as compared to healthy users [Séguin et al., 2019]. Indeed, BCIs based on imagined speech might not be suitable for all patients in the long term, such as those in which motor impairments are accompanied by progressive cognitive decline (e.g. LIS [Séguin et al., 2024]). They might however benefit many individuals in whom these functions are spared, for instance, patients with post-stroke aphasia, where attention and global control are generally preserved [Brownsett et al., 2014] and importantly, imagery skills are better retained than spoken language [Fama et al., 2017; Fama and Turkeltaub, 2020; Sierpowska et al., 2020; Stark et al., 2017]”

In addition, we have removed the following related claims from the abstract “[...] *non-invasive BCI-learning can help predict the individual benefit from an invasive speech BCI*”.

Last, it is important also to note that the goal of the present study is not to provide a new means of communication for speech disorders (as stated on page 23, lines 775-776: “*Although surface EEG-based BCIs are unlikely to become stand-alone communication devices [...]*), but to address “*speech-BCI controllability from a neurophysiological rather than neuroengineering perspective*” (page 3, lines 93-94 in the introduction). Accordingly, we believe that interpreting our findings exclusively from a clinical perspective as a tool for communication deviates from the primary, broader, basic neuroscience scope of our study.

As for the need for a control condition, we concur with the Reviewer about its relevance and we are pleased to include in the revised version of the manuscript, new results showing that providing accurate real-time feedback is necessary to allow users to acquire BCI-control.

We show that learning is disrupted in a separate group of participants trained to control the BCI with a discontinuous real-time feedback (see paragraph “*Experiment with discontinuous real-time feedback*”, page 6, lines 210-220 in the Methods section, Supplementary Methods, and Figure 1 below), even though the key decoding features overlap with those from the group trained with the continuous feedback (dataset presented in the original version of the manuscript).

Figure 1. Evolution over the training period of BCI-control performance in the group of participants who trained with a continuous feedback (green) and the group who trained with a discontinuous feedback

(violet). We observed a significant improvement selectively in the first group of participants, demonstrating that the real-time feedback is key to allowing users to acquire BCI-control.

We also provide behavioral data about the ability to perform the syllable imagery in both groups of participants. These data show differential dynamics across the training period related to BCI-control learning (see “*Mental Chronometry Results*”, page 19, lines 622-646 in the Results section).

Reviewer #2

1. What are the major claims of the paper?

A binary imagined speech BCI with EEG can improve with training across multiple days, though not for all participants, similar to the larger (hand) motor imagery literature. The improvement is linear. Theta (frontal and central) and gamma (left temporal) features are most related to the improvement in performance.

2. Are the claims novel? If not, please identify the major papers that compromise novelty. Will the paper be of interest to others in the field? Will the paper influence thinking in the field?

Yes, I think this paper is important as it highlights the power of training especially on the user-level. The study includes a lot of data, which is very valuable.

3. Are the claims convincing? If not, what further evidence is needed?

The claims are pretty convincing, though additional information would make it stronger.

The task section ‘syllable imagery’ should include the fact that participants were asked to do the imagery for 5 seconds continuously (?), and whether or not that meant repeating imagining saying the syllable many times.

Along the way, participants may have changed the strategy they used to control the BCI (imagine the syllable) towards a typical neurofeedback strategy, or anything that seemed to work to modulate their brain rhythms, for example. I would suggest the authors to include a report of what strategy the participants used during the experiment, retrospectively [...].

Answer. We would like to thank Reviewer 2 for the time taken to review our manuscript and the very useful and detailed feedback.

We understand that the description of the imagery task lacked clarity and we have provided more information in this regard in the revised version of the manuscript as follows: “*Participants were instructed to start imagining repeating each of the two syllables (such as “fo/-fo/-fo/...” or “gi/-gi/-gi/...”)* while keeping a constant pace.”

The importance of complying with the task’s instruction and keeping a consistent strategy for imagining the syllables across the 5 days of training is an important aspect and we now provide more information on this point. We have included a table (Supplementary Table 1) with participants’ final debrief describing the strategy they have used to control the BCI, together with other comments on internal states that can affect the performance (e.g. fatigue). From these reports, we can see that participants complied with imagining articulating the syllable. Interestingly, the single participant who reported a different strategy was the individual displaying the lowest (negative) learning slope.

In addition, decoding features (i.e. frequencies and brain regions) contributing the most to the classification of the two syllables remain similar across the 5 days of training, as indicated by Figure 2d (same as Figure 2 below). With due caution, we consider these results as indirect evidence for a consistent strategy across individuals and throughout training.

Figure 2. Maps visualizing the average feature weights across participants, for each training day. On each map, channels are represented on the y-axis, and individual frequency values are on the x-axis. Scalp topographies below each map display the average weight for each channel, across all frequencies and participants, on each training day.

Furthermore, exploring different strategies could likely be detrimental to the real-time BCI-control given the design of the BCI experiment itself. The visual feedback during the online session is based on the classifier computed during the preceding offline session, and thus optimal decoding is achieved by keeping a consistent strategy across sessions.

In addition, we now include in the revised manuscript data from a mental chronometry task performed before the BCI-control session on each day, that consisted of measuring the time required to repeat aloud versus imagining repeating each of the two syllables 5 times. This task is a well-known experimental approach to empirically evaluate imagery skills (see for instance [Guillot and Collet, 2005]) and has previously been used to quantify motor imagery abilities for BCI-control [Marchesotti et al., 2016]. According to this previous literature, the temporal congruency between imagery and execution indicates good imagery abilities.

As participants performed this test every day before the BCI-control session, we assume they were strongly primed to keep a consistent strategy in the imagery task. On average, participants took between 4 and 4.5 sec to repeat/imagine speaking the 5-syllable stream (see Figure 3 below) with a similar variance in both tasks.

Figure 3. Mental chronometry task. Time required to perform 5 repetitions of the syllable as used during the BCI-control task. The task is performed through imagery (pink) and by speaking aloud (yellow).

We also observed that participants became faster in both output modalities as training progressed: the difference between the 1st and last day was on average below half a second, with a difference of 460 msec for imagery and 430 msec for the speaking condition. Assuming that participants kept the same imagery rhythm during the mental chronometry task and the BCI experiment, the total number of repeated syllables during the 5 seconds of BCI control on the last day should be equal from day 1 to 5. It is thus unlikely that better decoding accuracy arose from a higher number of repeated syllables. On the contrary, we found a negative trend when correlating the average BCI-control performance with the syllabic rate measured with the mental chronometry (see below Figure 4, left), suggesting that participants with slower repetition were also those who achieved better control. A similar negative trend was found for the learning slope (see below Figure 4, right).

Figure 4. Relationship between BCI-control and syllabic rate during speech imagery. We considered the average BCI-control performance and learning slope across the 5 days of training and tested for a relationship with the syllabic rate during speech imagery as measured with the mental chronometry test.

[...] The EEG-EMG analysis did not include a test on the linear increase throughout training on the offline EMG data. This does seem to be significant so I would suggest the authors to report it along with some discussion on what this could indicate. For both offline and online analyses, they (EEG and EMG) do seem to follow the same trend. I also wonder if there was a difference between the responders and non-responders (did the responders have more motor activity perhaps?).

Answer. The Reviewer's intuition is correct in the sense that we do observe a statistically significant linear increase in the CV accuracy obtained by considering the EMG data from the *offline* session across the 5 days. We now report these results as follows in lines 590-593: "*Data from the EMG-offline session showed that the CV accuracy increased linearly across training ($F_{1,59} = 8.69, p = 0.0045, \eta^2_p = 0.13$).*"

We now document the linear increase in the CV accuracy obtained from both EMG and EEG data in both sessions (using the same approach as the one to compute the learning slope of the behavioral performance). We further extend the analysis of EMG signals by considering the slope of the CV Accuracy: we show that this was higher when decoding the EEG data during the online session than in the offline session and in EMG data (see Figure 5 below, left, same as Fig.4b in the manuscript). This suggests that while EMG signals likely contained information about the syllable choice, learning involved changes occurring at the level of neural activity elicited by covert speech. We report these results as follows in lines 17, lines 593-599:

“We further investigated differences in the evolution of the CV accuracy across training by computing the training slopes for the EMG data, in both sessions (Fig. 4b), as previously done for the EEG data (Fig. 1e). We found that overall, the slope of the CV accuracy was higher with the EEG data ($F_{1,14} = 4.5$, $p = 0.05$, $\eta^2_p = 0.24$) and that the interaction between Data and Session was nearly significant ($F_{1,14} = 3.36$, $p = 0.08$, $\eta^2_p = 0.19$). This latter effect was mainly driven by higher values of the learning slope in the EEG-online than EMG-online ($T_{14} = 2.45$, $p = 0.02$, $d = 0.63$), and EMG-offline data ($T_{14} = 1.78$, $p = 0.09$, $d=0.46$). There was no main effect of Session ($F_{1,14} = 0.57$, $p = 0.46$, $\eta^2_p = 0.039$)”

Figure 5. (left) Increase across the training period (slope) of the CV model accuracy obtained using EEG (solid pattern) and EMG (striped pattern) data. Results relative to the offline session are depicted in blue and those for the online session in salmon. **(right)** Correlation between the learning slope modeling the behavioral improvement in BCI-control (y-axis) and the CV Accuracy slope obtained considering the EMG data during the online session.

The link between the aptitude of learning to control BCI and EMG activity is a very interesting point to address. As the difference in sample size between good and poor learners (11 against 4) prevents us from doing group comparisons, we thus ran a correlation analysis between the learning slope obtained considering the BCI-control performance and the linear increase in the CV accuracy obtained with the EMG data during the online session. We found a marginal correlation close to statistical significance ($r = 0.5$, $p = 0.058$, see above Figure 5-right, same as Fig.4c) and report this result in the manuscript as follows (page 18, lines 600-602):

“Next, we tested for a relationship between the slope obtained considering the EMG-online dataset and the behavioral improvement in BCI-control (learning slope), and found a close to significant positive correlation ($r = 0.5$, $p = 0.058$, Fig. 4c)”.

We also extend the discussion regarding the EMG activity as follows :

“Given extended reports indicating residual EMG activity during inner speech and even the possibility of above-chance EMG decoding (see for a review [Perrone-Bertolotti et al., 2014]), we tested whether EMG signals alone could be classified by computing the CV accuracy in post-processing. The underlying hypothesis is that learning effects should be absent or at least less pronounced in EMG than in EEG data. Consistently, we found no decoding improvement during the online session with EMG, and no distinctive decoding features. However, the above-chance decoding on some of our EMG datasets and a significant increase in CV accuracy offline, indicate that speech imagery was likely accompanied by subthreshold motor activation, a finding that is compatible with the fact that participants were instructed to imagine pronouncing the syllables (rather than e.g. hearing syllables). The presence of residual EMG activity during mental imagery has been the subject of debate for almost a century [Guillot et al., 2012; Jacobson, 1932] and there are still contrasting results in the field of speech imagery [Kapur et al., 2018; Meltzner et al., 2008; Nalborczyk et al., 2020; Oppenheim and Dell, 2010; Perrone-Bertolotti et al., 2014]. Our results are in line with the Motor Simulation View (in contrast with the Abstraction View) of inner speech, where peripheral muscular activity would result from imperfect inhibition of motor commands [Guillot et al., 2012; Perrone-Bertolotti et al., 2014], possibly accounting

for the selectivity of EMG signals to specific phonemes [Livesay et al., 1996; Locke and Fehr, 1970; McGuigan and Dollins, 1989; Nalborczyk et al., 2017; Nalborczyk et al., 2020]. Above-chance decoding found on average on some days confirms that EMG activity is more than a merely non-specific tonic activation [Kapur et al., 2018; Nalborczyk et al., 2020], and is subject to marked inter-individual differences [Nalborczyk et al., 2020]. EMG activity during imagery is also modulated by the intensity of the mental effort [Guillot et al., 2012; Slade et al., 2002], an effect that given the participants' reported experience, likely contributes to the observed correlation with the learning slope. Critically, during the online session, EEG-based decoding achieved significantly higher accuracy than EMG-based decoding, and showed an improvement across training, unlike EMG-based decoding. This, along with the distinct patterns of decoding frequencies, indicates that potential contamination of EEG signals by EMG activity (either as muscle artifacts or neural activity elicited by overt speech) did not interfere with the acquisition of BCI-skills. While EMG signals likely contained information about the syllable choice, learning involved changes occurring at the level of neural activity elicited by covert speech. Significant learning effects might also be observed by providing a feedback based exclusively on EMG, given the high-decoding accuracy achieved with a speech-BCI based on EMG signals [Kapur et al., 2018]. This kind of closed-loop system could benefit patients who retain some residual orofacial movements, and for non-invasive solutions, might have some efficacy. This question remains however outside the scope of the present study”.

4. Are there other experiments that would strengthen the paper further? How much would they improve it, and how difficult are they likely to be?

A comparison experiment where the task instruction is modified could be an interesting control experiment. However, I don't think that is necessary for the current study to be valid.

Answer. We concur with the Reviewer that using different task instructions would be a great addition to these findings, as a control or further experimental conditions.

We now report a control experiment demonstrating that the temporal accuracy of real-time feedback is necessary to allow users to learn BCI-control. We trained a separate group of participants to control a BCI similar to the one used in the main experiment (i.e. same syllables, decoding approach, and training modalities) except that the real-time feedback was discontinuous (see paragraph “*Experiment with discontinuous real-time feedback*”, page 6, lines 210-220 in the Methods section, Supplementary Methods, and Figure 1 below). Feedback alteration consisted of visually informing the subject only when the decoded syllable was the cued one, thus displaying exclusively successful changes. In this group, learning was disrupted (see Figure 6 below), even though the decoding features were comparable with those found in the group trained with a continuous feedback.

Figure 6. Evolution over the training period of BCI-control performance in the group of participants who trained with a continuous feedback (green) and the group who trained with a discontinuous feedback (violet). We observed a significant improvement selectively in the first group of participants, demonstrating that the real-time feedback is key to allowing users to acquire BCI-control.

5. Are the claims appropriately discussed in the context of previous literature?

Yes.

6. Is the manuscript clearly written? If not, how could it be made more accessible?

The manuscript is clearly written for the most part. The introduction and methods could use a grammatical review (with a focus on the use of prepositions and plurality, specifics are in the annotated file). The second sentence of the abstract is very long, so I would suggest to split this up into separate sentences. The methods section was a bit difficult to read and understand on the first iteration. I would suggest to make it more concise, avoid repetition and be more clear in defining terms.

Answer. We regret the presence of grammatical errors and we thank the Reviewer for providing detailed annotations. We have now corrected them all and run a further edition check. We have done our best to make the methods section more concise.

We have shortened the abstract, modified the second sentence as suggested, integrated comment #8 from this Reviewer and Reviewer 1 regarding clinical implications, and referred to the main result of the control experiment.

7. Could the manuscript be shortened to aid communication of the most important findings?

See previous comment, the methods may be shortened.

Answer. We recognize the importance of conciseness and acknowledge that the methods are somewhat lengthy and following the reviewer's comment, we did our best to shorten it. Besides, in the revised version of the manuscript we provide new data and results (a control experiment, a behavioral test evaluating imagery skills, and EMG analyses) that inevitably lengthen the methods section. To minimize this, we propose to present part of this new material in the Supplementary Material section.

8. Have the authors done themselves justice without overselling their claims?

Yes, although the abstract can be toned down a bit. In line 26, it was stated that the findings indicate that learning 'must' be considered. It definitely shows it's a very important factor, but I do not see that the paper provides evidence that it's a 'must'. In the same sentence, it was stated that the findings indicate "that non-invasive BCI-learning can help predict the individual benefit from an invasive speech BCI". This statement would require a study on both EEG and invasive measurements, so I would not phrase this as an outcome of the current study, but rather a point for further research (for example, see Mansoureh Fahimi Hnazaee et al., 2022, J. Neural En; DOI: 10.1088/1741-2552/ac8764). Similar to how it was written in the actual conclusion section.

Answer. We thank the Reviewer for these relevant remarks. We agree that clinical work is required, and we have accordingly toned down the claims and rephrased the conclusions of the abstract as follows (page 1, lines 25-26):

"These findings demonstrate that combining machine and human learning is a successful strategy to enhance BCI controllability"

We also would like to thank the Reviewer for pointing out the study from Hnazaee and colleagues, which is highly relevant to our proposal of using a training based on a non-invasive BCI to identify candidates for invasive speech-BCIs. We have taken the remark of the Reviewer into account and removed the sentence "[...] non-invasive BCI-learning can help predict the individual benefit from an invasive speech BCI [...]" from the abstract and discuss this previous work in the conclusions as follows (page 23, lines 778-780):

Training to perform real-time control could be used as a benchmark to identify learners who would benefit most from intracortical BCIs. To validate such an approach, future work is required to 1- establish the correspondence between surface and intracranial EEG recordings of the neural activity elicited by imagined speech, as previously suggested for sensorimotor rhythms in patients with locked-in syndrome (LIS) [Hnazaee et al., 2022]

9. Have they been fair in their treatment of previous literature?

Yes.

10. Have they provided sufficient methodological detail that the experiments could be reproduced?

Yes.

11. Is the statistical analysis of the data sound?

As far as I can tell the statistical analysis is sound.

12. Should the authors be asked to provide further data or methodological information to help others replicate their work? (Such data might include source code for modelling studies, detailed protocols or mathematical derivations).

I would always suggest to provide a link to the scripts used for the analysis.

Answer: We have now provided the link below to a repository containing the scripts we used for the analysis:

<https://github.com/kinkinibhadra/Learning-to-operate-an-imagined-speech-Brain-Computer-Interface-...-neural-activity>

13. Are there any special ethical concerns arising from the use of animals or human subjects?

No, the human subjects appear to have been well informed.

14. Additional comments?

Please note the annotated manuscript for more detailed comments.

Answer. We thank the Reviewer for the detailed feedback. We have addressed below each of the comments.

Introduction

15. Page 2, line 37: The whole process is called a BCI, here would fit 'neural prosthesis' or 'device'. Please re-write.

Answer. We have rewritten the sentence by moving the term BCI at the end of the sentence.

16. Page 2, lines 73-74: Could use a reference or two (e.g.: Blabe C H, Gilja V, Chestek C A, Shenoy K V, Anderson K D and Henderson J M 2015 Assessment of brain-machine interfaces from the perspective of people with paralysis J. Neural. Eng.).

Answer. We thank the Reviewer for suggesting this interesting reference, which we have now cited in the manuscript (Introduction, page 2, line 77).

17. Page 2, line 77: This part is not clear to me, please rephrase ('imagined speech units'?)

Answer. We have rephrased the sentence as follows: "*Capitalizing on its far greater ease of use accessibility, several studies have employed surface EEG for decoding offline (i.e. open-loop) a wide variety of imagined speech units such as phonemes, syllables, and words"*

Materials and Methods

18. Page 4, lines 119-120: Why focus on kinesthetic sensation, if the target population is people who have speech impairment through damaged motor areas? I would like to see some additional rationale for this.

Answer. We agree with the reviewer that it is important to provide a rationale for instructing participants to focus on the kinesthetic sensation associated with imagining repeating the syllable, given that different kinds of imagery (e.g. auditory vs kinesthetic) might activate different brain regions. We were particularly interested in recruiting the network involved in speech production as this is more closely comparable with existing work in patients affected by disorders of speech production (e.g. anarthria, [Metzger et al., 2023; Moses et al., 2021]). In addition, kinesthetic speech imagery might activate more superficial regions in the inferior parietal lobe than imagined speech perception, likely activating structures less accessible with surface EEG such as the auditory cortex and the superior temporal gyrus [Ferryhough and Alderson-Day, 2015]. We have extended the motivation as follows (page 5, lines 167-171):

"As the long-term goal of speech-BCI is to provide a means of communication for individuals who have lost the ability to speak, and consistent with the latest works of speech-BCI [Metzger et al., 2023; Moses

et al., 2021], *participants were instructed to focus on speech production rather than imagine hearing oneself speaking or imagine the syllable written in characters. Kinesthetic motor imagery also recruits brain areas more superficial than imagined speech perception thus more accessible with surface EEG [Fernyhough and Alderson-Day, 2015].*"

19. Page 4, line 153: Did they have to repeat the syllable many times? As many times as possible? Please describe the specific instructions and, if you can get them, some report of what the participants did in the end?

Answer. We have addressed this point in our answer to comment #3.

20. Page 4, line 157: This is inconsistent, 10 trials per syllable per block?

Answer. We thank the reviewer for spotting this mistake, we have now corrected the text as follows: "*each consisting of ~~20~~ 10 trials per syllable*".

21. Page 5, line 180: So 10 trials per syllable per block, 20 trials per block, 80 trials in total?

Answer. Yes, this is correct.

22. Page 5, line 188: How can you calculate a percentage of trials within one trial? Remove the 'for each trial'?

Answer. We thank the reviewer for identifying this error, we have now corrected the sentence as follows:

"Participants' ~~performance in controlling the BCI (BCI-control performance)~~ was calculated by considering, for each trial, the ~~percentage of rate (%) of trials where~~ the classifier's outputs that corresponded to the cued syllable."

23. Page 5, line 191: Why assume a linear change?

Answer. We agree that learning dynamics can follow trends different from a linear one, however without any specific assumption and to ease the interpretability of our results, we opted for the easiest model.

24. Page 5, line 196: What is meant by learning dynamics? It may be good to define this term beforehand

Answer. We have provided a definition as follows: "*learning dynamics (i.e. the evolution of performance across the 5 training days)*".

25. Page 5, line 199: What is this data exactly?

Answer. We have modified the text as follows: "*the average BCI-control performance ~~during the entire training across the entire-whole~~ training period*".

26. Page 7, line 282: It would be good to have an overview of which channels were noisy and were not 'real' data

Answer: Noisy channels where those typically affected by poor contact between the EEG electrodes and the scalp (e.g. due to hair density), namely those over lower and posterior regions.

Results

27. Page 9, line 355-356: What if you separate the learners from the non-learners?

Answer. It is an interesting suggestion to investigate a link between the improvement in BCI-control and the difference in the slope calculated based on the online and offline CV accuracy. Given that the different sample size between good and poor learners (11 vs 4 individuals) prevented us from doing between groups analysis, we performed a correlation between the learning slope obtained from the BCI-control performance (behavioral variable) and the difference in the slope obtained by considering the evolution of the CV accuracy between online and offline sessions. We found a positive trend that however was not statistically significant ($r = 0.32$, $p=0.24$, see Figure 7 below).

Figure 7. Correlation between the learning slope modeling BCI-control performance improvement (x-axis) and the difference in the CV accuracy slope between the online and offline session (y-axis). Data points are colored according to the slope of the CV accuracy online, red for positive values and blue for negative ones).

28. Page 10, line 369-370: I don't understand the difference between c and e (online), if it's the same, remove c?

Answer. We have expanded the legend as follows: “(b) Average BCI online performance (%) over the 5 training days. BCI performance is computed, separately for each trial, as the percentage of the classifier outputs in accordance with the cued syllable. (e) Model cross-validation (CV) accuracy obtained by computing the classifier considering the entire dataset from ~~in~~ the offline (blue) and online (red) sessions on each day (left).”

Please note that the difference in panel indexing is due to changes in the revised version of Figure 1.

29. Page 10, line 368-369: It's the other way around in the figure.

Answer. We thank the Reviewer for catching this mistake that we have now corrected.

30. Page 10, line 370: Average across participants, error bars are SE? Please include this information.

Answer. We have now added more information about in this figure, as well as Fig.4 and Supplementary Figure 1 and as follows: “Boxes represent the interquartile range (IQR), with the horizontal line indicating the median, and whiskers extending to data points that are within 1.5× the IQR from the upper and lower quartile. Individual points represent data from a single participant, and gray lines connect data points from the same participant. [...] Error bars in (b) and (f) indicate the standard error of the mean.”

31. Page 10, line 376: p < 0.001 for consistency.

Answer. We now report exact p-values for all the statistics.

32. Page 13, line 406: Please add a colormap and I don't understand what the dots left of the title represent?

Answer. We thank the Reviewer for spotting that the colorbar was missing, we have now added it. The red dotted line refers to panel f), where it represents the euclidean distance for one feature (one frequency-channel pair), that in panel g) is represented for all features. We have now moved the line to panel f) as in figure 8 below and specified in the figure legend what this red dotted line represents.

Figure 8. Updated panels f) and g) of Figure 2, displaying a schematics of the approach used to calculate the global index and based on the Euclidean distance in the features' space.

33. Page 13, line 406: What is this global index exactly (please define)? From within a participant I assume, then what channel/frequency? Or averaged across all?

Answer. Indeed, the global index was mentioned in the legend and results section but was not properly defined in the Methods section. It now reads: “The four distance matrices (one for each couple of consecutive days) were then averaged to obtain an individual index per participant, referred hereafter as the global index” (page 8, lines 314-315) and in the legend as follows “Correlation between the global index representing the amount of change both in the frequency and spatial domains (computed as the Euclidean distance between the features' weight from two consecutive days) obtained with the euclidean distance approach and the average BCI performance across the 5 training days” (page 15, lines 496-498).

34. Page 13, lines 419-420: There seems to be a third range (26-30Hz), why didn't you include this range here nor as a brainplot?

Answer. The reviewer is right that there is another frequency interval (28-30Hz) wherein the decoding features exhibit a decrease across training. We opted not to discuss this interval due to its much smaller range as compared to the other two intervals. Nonetheless, we present it here for the reviewer's consideration (Figure 9 below).

Figure 9. Brain topography relative to the 28-30 Hz range. Black dots highlight electrodes whose contribution to the decoding shows a statistically significant change across training (p -value < 0.05, permutation test). Negative t-values values corresponds to a decrease in contribution, positive values an increase.

35. Page 15, lines 482-483: This is indeed important, but it fails to mention this test for offline data. There does seem to be a significant increase there, please report. And some discussion on what this could indicate? For both offline and online analyses, they (EEG and EMG) do seem to follow the same trend. I also wonder if there was a difference between the responders and non-responders (did the responders have more motor activity perhaps?). I would suggest to include these analyses.

Answer. We have addressed this point in the second part of comment 3 from the Reviewer. We have also substantially discussed EMG results in the discussion (pages 21-22, lines 705-731).

Discussion

36. Page 16, lines 512-513: What is meant by feedback rate exactly? The amount of feedback (information) or the speed, or something else? Please specify.

Answer. We understand that the reference was not clearly cited. We now cite more relevant studies on humans instead of non-human primates, which similar to our paradigm focus on training participants as follows (page 21, lines 672-674):

“Better learning with continuous than discontinuous real-time feedback aligns with previous observations made during motor-BCI training [Neuper et al., 1999; Roc et al., 2021] and indicates that accurate and high-rate feedback enables an optimal error-driven strategy to control the BCI”.

~~*“This difference highlights the relevance of providing real time feedback to the user to enhance the discriminability of neural patterns, and is in line with improved performance with a higher feedback rate as previously shown in non-human primates.”*~~

37. Page16, lines 521-522: However, you did instruct the participants to use a kinesthetic strategy, which should target the ventral motor cortex, which should be similarly superficial to hand motor imagery? Do you think they may have adapted their strategy?

Answer. The reviewer is right in pointing out that the ventral motor cortex, often reported in motor imagery of speech, is more superficial than other regions involved in speech imagery. The limitation is more related to lower the spatial separability of imagined speech units as compared to the interhemispheric lateralization in the imagery of the hand’s movement. Accordingly, we modified the sentence as follows (page 21, line 682): *“This is likely due to the relatively weak neural signals elicited by speech imagery, their limited spatial separability and the restricted access to deeper speech brain regions with surface EEG”.*

As for the strategy change across training, we discuss this point in detail in response to Reviewer's comment #3.

38. Page 19, line 617: Change to 'change', since 2-tailed (also decrease)
Answer. Corrected

Reviewer #3

The present study is devoted to further understanding how imagined speech could be decoded from surface EEG recordings using a BCI system that could translate these recorded signals into text or synthesized speech. This is an exciting field and many new advances have been produced in recent years. More in particular, decoding imaged speech could be important for helping patients in which speech production areas have been damaged (e.g., post-stroke aphasia; recent research have shown preserved inner speech in some of these patients that could benefit for this type of decoding techniques). In order to investigate more on this issue, the authors created a close-loop BCI system (relying on scalp EEG signals) that was proved to decode two imaged syllables (with clear phonetic dissimilarities that could maximize decoding capacity), in 15 participants, trained during 5 consecutive days. The experiment consisted in two phases, an offline experiment, where the authors collected data to train the classifier; and an online experiment, where they use the fitted classifier to predict the imagined speech. Their results show a slightly above chance accuracy considering the large interindividual variability, and certain amount of learning. Overall, although this is an important and well conducted experiment but many concerns arise when carefully reading this article.

Some major concerns about the study:

1. The main concern is related to the task used for the experiment. The task created in this study for imaged speech consisted in repeating an instructed syllable during 5 seconds, and this information is further used for training the BCI and further decoding. However, it is unclear how much repetitions are produced during this period internally, and it seems a very uncontrolled experimental situation (for example, variability might exist on how long where the syllables produced by each participant, where the participant following a particular rhythm, how many syllables produced each participant, etc.). As the imaged internal speech is not properly structured, it is difficult to correctly train the classifier (indeed, part of the EEG epoch might not correspond to the expected internal speech adding noise to the signal). For example, a better approach would have been to repeat only once each syllable (e.g., 1 s internal slow production), and therefore, ensuring that the whole EEG epoch would correspond to the internal generated signal (reducing noise of silence or non-production stages). As it is right now it is difficult to see how this uncontrolled setting could derive in a good classification pattern and further learning across days. A recent study (see Mor Regev and colleagues, in *Cer. Cortex*, "Mapping Specific Mental Content during Musical Imagery") tried a very interesting approach to study for first time music covert productions. They pre-trained participants with a particular melody (particular rhythm) and later in the scanner, the requested participants to mentally replay this melody internally. In this situation, the correlation analysis between the real signal and the internal production (fMRI analysis) allowed them to infer the quality of the signal reproduced internally. Similar settings could be designed to ensure a larger degree of control about the overt or internal speech signal. To my view, the design of the task was not sufficiently innovative and the authors might need to think more deeply on these important issues.

Answer. We thank the reviewer reviewing our work and for the useful comments. We fully agree on the importance of the behavioral task.

Regarding the innovative character of the task, we agree that syllables imagery is a rather classical one but we would like to emphasize that the goal of the experiment was to probe neural changes related to 5 training days, and not in the speech decoding performance.

Regarding the reviewer's doubts about the reliability of the task, we are pleased to address this matter with new data probing the regularity of the pace used by participants for repeating syllables. Before the BCI-control session on each day, participants performed a mental chronometry task, a behavioral test consisting of measuring the time taken to repeat aloud versus imagining repeating each syllable 5 times. This is a well-known experimental approach to empirically evaluate imagery skills (see for instance [Guillot and Collet, 2005]), which has previously been used to quantify motor imagery abilities for BCI-control [Marchesotti et al., 2016]. According to this previous literature, the smaller the delay between imagery and execution the better the imagery ability. We now provide the duration of 5 repetitions of the same two syllables used in the BCI task, for both the imagined and speaking modality. To evaluate innate inter-individual differences in the syllabic rate, we present below the data from the first day

showing that inter-individual variability is rather limited and consistent between the imagining and speaking modalities (Figure 10 below).

Figure 10: Mental chronometry task results during the 1st day of training. (left) Duration of 5 repetitions of the same two syllables for the imagined and speaking modality, with lines connecting data points from the same participant. (right) Table with average duration, standard deviation across participants, minimum and maximum duration of 5 syllables repetition for the two modalities (imagine and speaking).

In addition, it is reasonable to presume that participants, primed by performing the mental chronometry task at the beginning of the experimental day, likely maintained the same pace during BCI-control. If this was true, the repetition rate did not influence the BCI performance. Accordingly, we found only a weak – negative and not significant - relationship between the syllable repetition rate measured during the mental chronometry task and two measures from the BCI-control task, respectively the average BCI-control performance, and learning slope across training (see below, Figure 11).

Figure 11. Link between BCI-control and mental chronometry. We considered the average BCI-control performance and learning slope across the 5 days of training and tested for a relationship with the syllabic rate obtained from the mental chronometry test for the imagine and speaking modalities.

Regarding the choice of syllable repetition, we do not believe that a single syllable output would have improved decoding. On the contrary, when consistent as is the case here, repetition minimizes the notorious temporal alignment problem in speech-imagery decoding, consisting in the impossibility of determining with precision the onset of the imagery, even when participants are cued precisely on when to start imagining. A solution for this issue in offline analysis is generally the use of dynamic time warping as previously proposed for decoding imagined speech from intracranial signals [Angrick et al., 2021; Martin et al., 2014]. However, the limited spatial resolution of EEG limits decoding accuracy and the possibility of applying this kind of realignment solution.

We agree that pre-training participants to a particular rhythm as in Regev et al. is a good experimental choice, and precisely we believe that the mental chronometry task administered prior to the BCI-control phase has primed participants to keep consistency in the imagery task, similar to the method used by Regev et al.

2. Authors used a random forest procedure to decode the syllable imagery. Random forests could be a good option for the offline part, since they are fast to train, but there are slow to test (the multiple decision trees have to predict the label of the test data). Therefore, these classifiers are not the best ones to perform real-time predictions. In fact, authors did not report the delay between data collection, in the online experiment, and the given feedback.

Answer. We agree that Random Forest is not the optimal choice for decoding speech imagery in real-time. For instance, recurrent neural networks might offer better decoding performance, or adaptive classifiers, that are the object of another line of research from our group (see Wu et al., 2024). RF remains however a very useful approach for feature selection.

The reason to use RF here was that it enables a fairly straightforward analysis of the decoding features computed offline and the monitoring over time.

We thank the reviewer for pointing out that some information was missing, that we have now added in the manuscript as follows (page 5, lines 203-204):

“The delay between the recorded data and the feedback presentation was on average 100 ms.”

Based on this, we can confidently consider the delay was negligible and did not hamper the real-time feedback presentation.

3. Authors considered the weights of the first 200 features, based on the cumulative sum, but looking at Suppl. Fig 1, the elbow (around 75% of the cumulative sum) is found around 400 features.

Answer. The Reviewer is correct in identifying the elbow of the cumulative sum at higher features ranking (more specifically at a feature ranking around 500) where the cumulative sum reaches ~ 77%. We agree that considering a higher number of features could also be legitimate. However, we based the decision of considering the first 200 features not only on the cumulative sum but also on the standard deviation calculated across days of the cumulative sum (see bottom plot of Supplementary Fig. 2, previously Supplementary Fig.1) which reaches its peak around a feature rank of 170. Although this information is mentioned in the caption of the Supplementary Fig.2, we now include it in the main text, (page 7, lines 275-277), as follows:

“~~The choice of~~ This subset size was chosen based on ~~motivated by the fact that~~ the cumulative sum of the first 200 features’ weights exceeding on average 50% and on its standard deviation across training days increasing up to the 170th ranking place (Supplementary Figure. 2a). This shows that only a part of higher ranking features is most prominently affected by training”.

Furthermore, as features are ranked according to their weight in descending order, lower-ranking features are associated with a negligible weight. This can be seen in the plot below, comparing the original plot in Supplementary Figure 1d obtained with the first 200 features (see below, Figure 12 left), and the same plot considering the first 560 features (see below, Figure 12 right). From this, we ascertain that there is no significant difference in the trend across training.

Figure 12. The evolution across training of the features’ weight sum does not change when considering the first 200 (left) and first 560 features (right). This latter value was chosen considering the elbow of the cumulative sum in Supplementary Fig. 2.

4. As authors stated, there is an increase of the CV accuracy across sessions, but the accuracy is surprisingly low. Indeed, it is around chance level (chance = 50%) in almost all offline sessions (blue line in Fig. 1E). As the authors discussed, the CV accuracy improved in the online session, but this accuracy is the result of an offline classification of the online data. With these results, it is not surprising the low performance of the BCI system (Fig. 1C). In fact, something that I wonder is whether the participants were upset with the system because it continuously failed. Did the authors asked participants for feedback? This is another aspect that need to be

improved, asking participants to provide a report of their internal production as well as opinions about the experiment in general.

Answer. This is evidently a crucial point, and indeed the limited BCI-control impacts on users' experience and motivation. We expected classification accuracy to be low, given the use of a surface EEG system and based on previous studies (see for instance [Sereshkeh et al., 2017]). For this reason, we verbally informed participants that they might feel some frustration due to poor BCI-control and instructed them to keep focusing on the imagery task despite any potential frustration. To compensate for this, the monetary bonus served the purpose of keeping participants motivated. Importantly, by providing the bonus when performance exceeded that of the previous day, we aimed at mitigating the effect of overall poor performance and provided an attainable objective.

As for participants' feedback, we collected their subjective reports about the strategy used and general impressions. We now provide the individual responses in a separate table (Supplementary Table 1) in the updated version of the manuscript. Participants mentioned that fatigue and lack of motivation affected the control. Of note, the order of participants' responses in the table is based on the learning slope, in descending order.

5. Another surprising result is the EMG accuracy (Fig. 4), which is very similar to EEG accuracy in the offline experiment, and it is slightly lower in the online experiment. The EMG pipeline considered just 70 features, while EEG pipeline considered 2,135 (30 times more features than EMG). I think that these results should be more detailed in the discussion section.

Answer. Although we understand this comment, we believe that the different number of features does not play a key role in the results because the most discriminant features are qualitatively different in the two modalities (see below, Figure 13).

Figure 13. Average features' weight across all experimental days considering a classifier obtained considering exclusively EEG data (left) and EMG data (right). Colors indicate the different frequency bands (theta: orange, alpha: green, beta: pink, low-gamma: blue, high-gamma: magenta).

In addition, we believe the difference between the EEG and EMG decoding accuracy during the online session is crucial, not only because on average the decoding is lower with the EMG data ($T_{14} = 2.77$, $p < 0.05$, $d = 0.71$), but importantly due to the difference in its evolution over training. As we reported in the manuscript “[...] while EEG-online accuracy showed a strong linear increase throughout training ($F_{1,59} = 17.79$, $p < 0.001$, $\eta^2_p = 0.23$), the same analysis performed with EMG-online data revealed no statistically significant change ($F_{1,59} = 2.53$, $p > 0.05$, $\eta^2_p = 0.04$)”.

Similarly, our new analysis on the learning slope of CV accuracy shows that only the EEG-online dataset presents a substantial linear increase (Figure 4c).

More generally, it is not surprising that it was possible to achieve above-chance decoding with the EMG accuracy given that EMG activity can reflect speech units such as phonemes during speech imagery [Livesay et al., 1996; Locke and Fehr, 1970; McGuigan and Dollins, 1989; Nalborczyk et al., 2017; Nalborczyk et al., 2020]. This however does not detract from the validity of the neural data, which are expected to be used in invasive speech-BCI procedures. We extended the EMG discussion as follows:

“Given extended reports indicating residual EMG activity during inner speech and even the possibility of above-chance EMG decoding (see for a review [Perrone-Bertolotti et al., 2014]), we tested whether EMG signals alone could be classified by computing the CV accuracy in post-processing. The underlying hypothesis is that learning effects should be absent or at least less pronounced in EMG than in EEG data. Consistently, we found no decoding improvement during the online session with EMG, and no distinctive decoding features. However, the above-chance decoding on some of our EMG datasets and a significant increase in CV accuracy offline, indicate that speech imagery was likely accompanied by subthreshold motor activation, a finding that is compatible with the fact that participants were instructed to imagine pronouncing the syllables (rather than e.g. hearing syllables). The presence of residual EMG activity during mental imagery has been the subject of debate for almost a century [Guillot et al., 2012; Jacobson, 1932] and there are still contrasting results in the field of speech imagery [Kapur et al., 2018; Meltzner et al., 2008; Nalborczyk et al., 2020; Oppenheim and Dell, 2010; Perrone-Bertolotti et al., 2014]. Our results are in line with the Motor Simulation View (in contrast with the Abstraction View) of inner speech, where peripheral muscular activity would result from imperfect inhibition of motor commands [Guillot et al., 2012; Perrone-Bertolotti et al., 2014], possibly accounting for the selectivity of EMG signals to specific phonemes [Livesay et al., 1996; Locke and Fehr, 1970; McGuigan and Dollins, 1989; Nalborczyk et al., 2017; Nalborczyk et al., 2020]. Above-chance decoding found on average on some days confirms that EMG activity is more than a merely non-specific tonic activation [Kapur et al., 2018; Nalborczyk et al., 2020], and is subject to marked inter-individual differences [Nalborczyk et al., 2020]. EMG activity during imagery is also modulated by the intensity of the mental effort [Guillot et al., 2012; Slade et al., 2002], an effect that given the participants’ reported experience, likely contributes to the observed correlation with the learning slope. Critically, during the online session, EEG-based decoding achieved significantly higher accuracy than EMG-based decoding, and showed an improvement across training, unlike EMG-based decoding. This, along with the distinct patterns of decoding frequencies, indicates that potential contamination of EEG signals by EMG activity (either as muscle artifacts or neural activity elicited by overt speech) did not interfere with the acquisition of BCI-skills. While EMG signals likely contained information about the syllable choice, learning involved changes occurring at the level of neural activity elicited by covert speech. Significant learning effects might also be observed by providing a feedback based exclusively on EMG, given the high-decoding accuracy achieved with a speech-BCI based on EMG signals [Kapur et al., 2018]. This kind of closed-loop system could benefit patients who retain some residual orofacial movements, and for non-invasive solutions, might have some efficacy. This question remains however outside the scope of the present study”.

Some minor concerns about the study:

1. There is a lack of machine learning terminology. For instance, the authors stated “classifier calibration”, while classifiers are usually “fitted” or “trained”.

Answer. We agree that the term “*calibration*” is less commonly used than “*training*” in reference to a classifier. We purposefully chose to use “*calibration*” to prevent confusion as we use the verb “*train*” to refer to the experimental design to train participants over 5 days. We also think the term “*calibration*” more clearly convey the concept that the classifier is re-trained on each day. Last, “*classifier calibration*” is used in the machine learning literature (Silva Filho, T., Song, H., Perello-Nieto, M., Santos-Rodriguez, R., Kull, M. and Flach, P., 2023. “*Classifier calibration: a survey on how to assess and improve predicted class probabilities*”. Machine Learning, pp.1-50, [Silva Filho et al., 2023].

We would appreciate the opportunity to address any remaining inaccuracies in machine learning terminology and make the necessary changes if the reviewer could provide specific examples of terms requiring refinement.

2. The battery in the offline experiment is increasingly fulfilled, but it is not specified how this battery is fulfilled in the online experiment (I guess it is binary: fulfilled or empty).

Answer. During the online experiment, the battery progressively filled or emptied based on the classifier output at each time sample, and not in binary mode. In this way, participants were presented with a continuous feedback, which we believe was crucial to allow them to learn the BCI-control. We have modified the methods as follows (page 5, lines 200-202):

*“[...] provided a **continuous** real-time feedback to the user. [...] The mapping of the decoder output to the battery feedback at each time sample was done in such a way that if the probability output by the classifier **changed in the direction of matched** the cued syllable, the battery's ~~would filling would~~ **increase, otherwise and it would decrease empty-if didn't.**”*

3. In the test for the linear increase in CV accuracy (using LMMs), authors could consider the interaction between planned contrast and sessions, instead of building separated models up.

Answer. This choice was motivated by our will to quantify the difference in the linear trend between the *offline* and *online* sessions. In the revised version of the manuscript, we present new CV accuracy data from the control group. We kindly direct the reviewer to refer to the updated results presented in the paragraph “*Training improves BCI-control abilities and decoding accuracy*” (page 10-11, lines 407-428).

4. A detailed rationale of the global index is necessary.

Answer. The rationale behind employing a global index was to derive a single value for each participant reflecting the change in decoding features throughout the training process. This approach enabled us to assess the correlation with BCI-control performance calculated across training sessions. We edited the Materials and Methods section to highlight the *consecutio-temporum* (page 15, line 517): “*We then investigated the link between the change in BCI-control performance and the change in the feature space. **To do so**, we extracted, separately for each participant, a global index representing the amount of change both in the frequency and spatial domains, computed as the Euclidean distance between the weight of two consecutive days.*”

Following another comment, we further extended the caption of Figure 2h as follows: “**(h) Correlation between the global index *representing the amount of change both in the frequency and spatial domains (computed as the Euclidean distance between the features' weight from two consecutive days) [...]***” (page 15, lines 496-498).

5. Report the exact p values of statistics.

Answer: We now report exact p-values throughout the entire manuscript.

6. Highlight the chance level in Figure 1C.

Answer: Done (now Figure 1b).

7. Make labels in Fig. 2 larger, please.

Answer: Done.

8. Write the full references in the reference section, please.

Answer: We believe the references in the manuscript comply with the standard Nature referencing style required by Communication Biology.

References

- Angrick M, Ottenhoff MC, Diener L, Ivucic D, Ivucic G, Goulis S, Saal J, Colon AJ, Wagner L, Krusienski DJ, Kubben PL, Schultz T, Herff C (2021): Real-time synthesis of imagined speech processes from minimally invasive recordings of neural activity. *Commun Biol* 4:1–10. <http://dx.doi.org/10.1038/s42003-021-02578-0>.
- Brownsett SLE, Warren JE, Geranmayeh F, Woodhead Z, Leech R, Wise RJS (2014): Cognitive control and its impact on recovery from aphasic stroke. *Brain* 137:242–254.
- Fama ME, Hayward W, Snider SF, Friedman RB, Turkeltaub PE (2017): Subjective experience of inner speech in aphasia: Preliminary behavioral relationships and neural correlates. *Brain Lang* 164:32–42. <http://dx.doi.org/10.1016/j.bandl.2016.09.009>.
- Fama ME, Turkeltaub PE (2020): Inner speech in aphasia: Current evidence, clinical implications, and

- future directions. *Am J Speech-Language Pathol* 29:560–573.
- Fernyhough C, Alderson-Day B (2015): Inner Speech: Development, Cognitive Functions, Phenomenology, and Neurobiology. *Psychol Bull* 141:931–965.
- Guillot A, Collet C (2005): Duration of mentally simulated movement: a review. *J Mot Behav* 37:10–20.
- Guillot A, Di Rienzo F, Macintyre T, Moran A, Collet C (2012): Imagining is not doing but involves specific motor commands: A review of experimental data related to motor inhibition. *Front Hum Neurosci* 6:1–22.
- Hnazaee MF, Verwoert M, Freudenburg Z V., van der Salm SMA, Aarnoutse EJ, Leinders S, Van Hulle MM, Ramsey NF, Vansteensel MJ (2022): Towards predicting ECoG-BCI performance: assessing the potential of scalp-EEG *. *J Neural Eng* 19.
- Höhne J, Holz E, Staiger-Sälzer P, Müller KR, Kübler A, Tangermann M (2014): Motor imagery for severely motor-impaired patients: Evidence for brain-computer interfacing as superior control solution. *PLoS One* 9.
- Jacobson E (1932): Electrophysiology of Mental Activities. *Am J Psychol* 44:677.
- Kapur A, Kapur S, Maes P (2018): AlterEgo: A personalized wearable silent speech interface. *Int Conf Intell User Interfaces, Proc IUI*:43–53.
- Lejko N, Larabi DI, Herrmann CS, Aleman A, Ćurčić-Blake B (2020): Alpha Power and Functional Connectivity in Cognitive Decline: A Systematic Review and Meta-Analysis. *J Alzheimer's Dis* 78:1047–1088.
- Livesay JR, Liebke AW, Samaras M, Stanley A (1996): Covert speech behavior during a silent language recitation task. *Biofeedback Self Regul* 21:381.
- Locke JL, Fehr FS (1970): Subvocal rehearsal as a form of speech. *J Verbal Learning Verbal Behav* 9:495–498.
- Marchesotti S, Bassolino M, Serino A, Bleuler H, Blanke O (2016): Quantifying the role of motor imagery in brain-machine interfaces. *Sci Rep* 6.
- Martin S, Brunner P, Holdgraf C, Heinze HJ, Crone NE, Rieger J, Schalk G, Knight RT, Pasley BN (2014): Decoding spectrotemporal features of overt and covert speech from the human cortex. *Front Neuroeng* 7:1–15.
- McGuigan FJ, Dollins AB (1989): Patterns of covert speech behavior and phonetic coding. *Pavlov J Biol Sci* 24:19–26.
- Meltzner GS, Sroka J, Heaton JT, Gilmore LD, Colby G, Roy S, Chen N, De Luca CJ (2008): Speech recognition for vocalized and subvocal modes of production using surface EMG signals from the neck and face. *Proc Annu Conf Int Speech Commun Assoc INTERSPEECH*:2667–2670.
- Metzger SL, Littlejohn KT, Silva AB, Moses DA, Seaton MP, Wang R, Dougherty ME, Liu JR, Wu P, Berger MA, Zhuravleva I, Tu-Chan A, Ganguly K, Anumanchipalli GK, Chang EF (2023): A high-performance neuroprosthesis for speech decoding and avatar control. *Nature*. Springer US. Vol. 620. <http://www.ncbi.nlm.nih.gov/pubmed/37612505>.
- Moses DA, Metzger SL, Liu JR, Anumanchipalli GK, Makin JG, Sun PF, Chartier J, Dougherty ME, Liu PM, Abrams GM, Tu-Chan A, Ganguly K, Chang EF (2021): Neuroprosthesis for Decoding Speech in a Paralyzed Person with Anarthria. *N Engl J Med* 385:217–227. <http://www.ncbi.nlm.nih.gov/pubmed/34260835>.
- Murray LL (2012): Attention and Other Cognitive Deficits in Aphasia: Presence and Relation to Language and Communication Measures. *Am J Speech-Language Pathol* 21:S51–S64.
- Nalborczyk L, Perrone-Bertolotti M, Baeyens C, Grandchamp R, Polosan M, Spinelli E, Koster EHW, Løevenbruck H (2017): Orofacial electromyographic correlates of induced verbal rumination. *Biol Psychol* 127:53–63.
- Nalborczyk L, Grandchamp R, Koster EHW, Perrone-Bertolotti M, Løevenbruck H (2020): Can we decode phonetic features in inner speech using surface electromyography? *PLoS One* 15:1–27.
- Neuper C, Schlögl A, Pfurtscheller G (1999): Enhancement of Left-Right Sensorimotor EEG Differences During Feedback-Regulated Motor Imagery. *J Clin Neurophysiol* 16:373–382.
- Oppenheim GM, Dell GS (2010): Motor movement matters: The flexible abstractness of inner speech. *Mem Cogn* 38:1147–1160.
- Perrone-Bertolotti M, Rapin L, Lachaux JP, Baciú M, Løevenbruck H (2014): What is that little voice inside my head? Inner speech phenomenology, its role in cognitive performance, and its relation to self-monitoring. *Behav Brain Res* 261:220–239. <http://dx.doi.org/10.1016/j.bbr.2013.12.034>.
- Roc A, Pillette L, Mladenovic J, Benaroch C, N'Kaoua B, Jeunet C, Lotte F (2021): A review of user training methods in brain computer interfaces based on mental tasks. *J Neural Eng* 18.
- Séguin P, Maby E, Fouillen M, Otman A, Luauté J, Giroux P, Morlet D, Mattout J (2024): The challenge of controlling an auditory BCI in the case of severe motor disability. *J Neuroeng*

- Rehabil 21:1–15.
<https://www.medrxiv.org/content/10.1101/2023.01.10.23284295v1><https://www.medrxiv.org/content/10.1101/2023.01.10.23284295v1.abstract>.
- Séguin P, Maby E, Mattout J (2019): Why BCIs work poorly with the patients who need them the most? In: . Proceedings of the 8th Graz Brain-Computer Interface Conference 2019.
- Sereshkeh AR, Trott R, Bricout A, Chau T (2017): Online EEG Classification of Covert Speech for Brain-Computer Interfacing. *Int J Neural Syst* 27:1–16.
- Sierpowska J, León-Cabrera P, Camins À, Juncadella M, Gabarrós A, Rodríguez-Fornells A (2020): The black box of global aphasia: Neuroanatomical underpinnings of remission from acute global aphasia with preserved inner language function. *Cortex* 130:340–350.
- Silva Filho T, Song H, Perello-Nieto M, Santos-Rodríguez R, Kull M, Flach P (2023): Classifier calibration: a survey on how to assess and improve predicted class probabilities. *Machine Learning*. Springer US. Vol. 112. <https://doi.org/10.1007/s10994-023-06336-7>.
- Slade JM, Landers DM, Martin PE (2002): Muscular activity during real and imagined movements: A test of inflow explanations. *J Sport Exerc Psychol* 24.
- Stark BC, Geva S, Warburton EA (2017): Inner Speech ' s Relationship With Overt Speech in Poststroke Aphasia. *J Speech, Lang Hear Res* 60:2406–2415.
- Willett F, Kunz E, Fan C, Avansino D, Wilson G, Choi EY, Kamdar F, Hochberg LR, Druckmann S, Shenoy K V, Henderson JM (2023): A high-performance speech neuroprosthesis. *Nature* 620:1031–1036.
<http://www.ncbi.nlm.nih.gov/pubmed/36711591><http://www.pubmedcentral.nih.gov/articlerender.fcgi?artid=PMC9882398>.
- Wu S, Bhadra K, Giraud A, Marchesotti S (2024): Adaptive LDA classifier enhances real-time control of an EEG Brain-computer interface for imagined-speech decoding. *Brain Sci* 14:1–21.

We would like to begin by thanking the Editor and Reviewers for reconsidering our revised manuscript. We greatly appreciate this opportunity, the positive feedback, and the additional comments. In the point-by-point responses below, we have addressed all the reviewers' comments. The **original comments from the reviewers** appear in bold, *text from the previous version* of the manuscript is in italics, and *newly added text* is in blue italics.

Reviewer #1 (Remarks to the Author):

The authors changed their manuscript according to the reviewer's criticism and included data of a control experiment demonstrating the improvement in performance of the imagery BCI. They should check their citations again, there are several (minor) errors:

**Metzger 2023 lacks volume info
Saroush et al wrong Vol
Maiseli et al lacks page info
Panachakel et al Journal missing
Panachakel 20221 all info missing
Nguyen et al lacks pages
Steyrl et al spelling mistakes
Ostenveld et al pages**

Answer. We would like to thank the reviewer for agreeing to review once more our manuscript and for their positive responses. We also thank the reviewer for catching the errors in the references, which we have now corrected.

Reviewer #2 (Remarks to the Author):

The authors made major revisions to the manuscript and added additional control experiments, for which I commend them. Unfortunately, the manuscript is still rather convoluted and lacking in logic, therefore not yet on par with the quality that I would have expected.

Following are a few more specific comments with regard to the rebuttal of my previous comments.

- 1. I appreciate the mental chronometry inclusion and additional information regarding the mental strategy participants used. The authors mention in the first comment that participants imagined the syllables 5 times in roughly a 5-second period for the mental chronometry. Was it also the instruction during the syllable imagery, to repeat it 5 times, approximately 1 syllable per second? This information should be added to the manuscript, for both tasks I don't see the *5* repetitions mentioned in either of the tasks. For clarity, I would also suggest to move the syllable imagery task explanation from the 'Offline session and classifier calibration.' section back to its own section underneath the experimental paradigm and mental chronometry sections.**

Answer. We would like to warmly thank the reviewer for taking the time to re-assess our manuscript and for these additional insightful comments.

The specific instruction to repeat the syllables five times (either through overt or covert speech) was limited to the mental chronometry task. This information was added to the revised version (page 4, line 134: "*To do so, we asked our participants to either repeat aloud (i.e. overt) or imagine pronouncing (i.e. covert) five times one of two syllables used for the BCI control*"). In contrast, during the syllable imagery used to control the BCI in the main experiment, participants were not instructed to imagine a specific number of syllable repetitions.

Following this comment, we've realized it was important to specify this experimental difference between the mental chronometry and the BCI-control, and added in the newly revised version of the manuscript as follows (page 5, lines 168-170): "*Unlike the mental chronometry test, they were not instructed to imagine a specific number of syllable repetitions, and the experimenter made no reference to this previous cognitive task.*"

In both the mental chronometry task and the BCI-control session, participants were allowed to proceed at their own preferred pace and were not required to maintain a consistent pace across the two tasks (mental chronometry and BCI-control).

As suggested by the reviewer, we have moved the description of the syllable imagery task during BCI-control to a separate paragraph, in addition to the one already present at the beginning of the Material and Methods section (page 3, from line 108: “*Experimental paradigm and syllables imagery*”).

2. In the first comment, it was also mentioned that “Interestingly, the single participant who reported a different strategy was the individual displaying the lowest (negative) learning slope.” I disagree with this statement, as this participant also reported imagined articulation and other participants reported multiple additional strategies. A better argument would be the (trending) negative correlations between performance and syllable imagery rate, perhaps taking individual data points rather than the average of 5 days can aid in the correlation.

Answer. The reviewer is correct in pointing out that the participant reported having complied with the instruction of performing kinesthetic imagery, but mentioned controlling the BCI using breathing. Concerning the negative correlation, the intuition of the reviewer is correct that including data from individual days (Figure 1b-c) resulted in a stronger relationship than in the original analysis where we considered the average across the 5 days (Figure 1a). We present below this new analysis, in two forms: with different colors for each participant (Figure 1b) and with different colors for each training day, together with the corresponding regression line (Figure 1c). From these new plots, we can observe a moderate trend towards a more negative relationship between syllable rate and BCI-performance as training progresses (Figure 1c), and that two participants with high BCI-control performance prominently drive the correlation (Figure 1b, white and blue-purple dots). The existence of this negative trend is surely interesting, however, we believe it is not sufficiently robust to be presented in the manuscript. We find it difficult to provide a mechanistic interpretation of this result, as the primary purpose was to compare the syllable rate during overt and covert production in order to assess the isochrony index.

Figure 1. Link between BCI-performance and syllabic rate (5 repetitions during the imagery task at the mental chronometry test). In (a) we considered the average across the 5 training days for each participant. (b) Each point corresponds to a participant on one experimental day (one color per participant). (c) Same as (b), but colors indicate different experimental days, with their corresponding regression line. The gray and dotted line is the same regression line as in (a) and (b), which is obtained by considering all data points.

3. I am satisfied with the additional tests and discussion on the EMG data.

Answer. We thank the reviewer for this positive feedback and for having appreciated the new results on EMG data.

4. A control experiment was added in which the “the real-time feedback was discontinuous, i.e. not systematically related to the classifier output and displaying only positive changes”, the reasoning for this was explained in the supplementary material. However, there was no explanation given for the addition of an “auditory cue similar to the sound of a metronome, imposing a pace for the syllable repetition arbitrarily set at 1.4 Hz”. Could this have anything to do with the odd finding in figure 5 that the first continuous feedback group got faster over time, whereas the new discontinuous feedback group slowed down over time?

Answer. We appreciate the comment and the interesting suggestion that this cue could induce a decrease in syllable rates during the mental chronometry task in the group trained with discontinuous feedback. However, we believe this is unlikely for two key reasons.

First, the mental chronometry task was performed without the auditory cue, and prior to the BCI-control phase, which reduces the likelihood that the auditory cue influenced the performance of the mental chronometry.

Second, as shown in Figure 2, the syllabic rate for the group that trained with discontinuous feedback decreased below the rate of the auditory cue (1.4 Hz), dropping on average from 1.08 to 0.94 Hz. If the auditory cue had influenced the syllabic rate, we would expect participants to increase their syllabic rate to adopt the cue's rhythm, rather than the opposite behavior.

We hope that this reasoning convinces the reviewer.

Figure 2. Syllable rate measured during the mental chronometry task. Number of syllables per second (Hz) obtained by considering the time required to repeat 5 times one of the two syllables for the imagery (magenta) and speaking (yellow) modalities in the group with discontinuous (darker colors) and continuous feedback (lighter colors) over the 5 training days.

5. The authors provided additional rationale for a focus on kinesthetic sensation of the current paper in the methods section. However, this doesn't take away my concern that it goes in stark contrast to what was written in the introduction: "Recent years have seen great advances in the field of speech-BCIs, most often through the decoding of motor representations of vocal tract movements from intracranial electrophysiological recordings 3–8, which have led to impressive decoding speeds reaching about 78 words per minute 4. Such an approach, however, is unlikely to be suitable for disorders of language where speech production areas are damaged, such as in post-stroke aphasia. A BCI appropriate for these disorders would require decoding representations of speech units produced through imagined, rather than attempted speech, in particular involving the language temporo-frontal system".

Answer. We agree with the reviewer in that the imagery task used to operate the BCI needs to be carefully chosen, in particular will have to be adapted to each individual's residual speech ability and specific impairment. We have added this last sentence in the discussion as follows (page 23, lines 789-792):

"In these patients, rehabilitative interventions based on closing the loop on imagery attempts with real-time feedback could be expected to mobilize residual neural patterns and promote neural plasticity. Importantly, such interventions will have to be adapted to each individual's residual speech ability and specific impairment".

Kinesthetic speech imagery remains a variant of imagined speech, upstream to sub-articulation, and could hence be suitable for some patients with post-stroke expressive aphasia. In the introduction, we have clarified this point and shortened the section as follows (page 2, lines 43-51):

“Recent years have seen great advances in the field of speech-BCIs, most often through the decoding of ~~motor representations of vocal tract movements~~ attempted speech from intracranial electrophysiological recordings¹⁻⁷, which have led to impressive decoding speeds reaching about 78 words per minute². Such an approach, however, is unlikely to be suitable for disorders of language where speech production areas are damaged, such as in post-stroke expressive aphasia. A BCI more appropriate BCI for these disorders would require decoding ~~representations of speech units produced through imagined~~, rather than attempted speech, in particular involving the language temporo-frontal system. Also termed covert or inner speech, imagined speech consists of the internal ~~pronunciation production of speech~~ without self-generated audible output^{8,9}, thus without the involvement of the musculoskeletal system. Depending on the brain damage location, different imagined speech strategies can be considered, from kinesthetic to abstract phonological ones. In ~~patients with aphasia, both subjective and objective assessments indicate that imagined speech is better preserved than spoken language, even in presence of severe overt (i.e. articulated and audible) speech deficits. Thus, investigating imagined speech has important implications for patients with aphasia, while remaining a valid approach for other disorders of speech production.~~”

Last, in our study, it was crucial to provide specific instructions to participants to ensure that their cognitive strategy was consistent across the entire group, particularly important for the EEG analyses. We have added this sentence in the Materials and Methods as follows (pages 3-4, lines 119-126):

*“Participants were asked to focus on the kinesthetic sensation they would experience if they pronounced the syllable aloud. As the long-term goal of speech-BCI is to provide a means of communication for individuals who have lost the ability to speak, and consistent with the latest works of speech-BCI^{4,52}, participants were instructed to focus on how they would articulate speech rather than how speech would sound or look like in writing ~~imagine hearing oneself speaking or imagine the syllable written in characters. Using imagined articulating speech, we expected to obtain a consistent neural response across the entire group and thus get reliable EEG analyses. Another technical advantage of exploring first this strategy is that kinesthetic imagery recruits more superficial brain areas than imagined speech perception thus more accessible with surface EEG~~*¹⁰.”

Additionally, the authors added a large section on deviation from isochrony index of the chronometry task of which I do not see the added benefit considering the already lengthy manuscript. Furthermore, the authors included many correlations of which they do not report the r and p values in the figures themselves and they must be interpreted very cautiously considering the very small sample size (N=15).

Answer. We agree with the reviewer that the results relative to the isochrony index are not very relevant given the absence of statistical significance. Based on this, we have decided to move the correlation plot in Figure 5 to the supplementary material, together with adding the r- and p-values as suggested by the reviewer in the figure below (Figure 5). In addition, we have added the r- and p-values also to Figures 1, 2, and 4.

Figure 5. Correlation between isochrony and behavioral BCI-control measures now appears in the supplementary material.

Reviewer #3 (Remarks to the Author):

The authors did a great job revising the comments I made, also the responses in general were very well developed. I do not have further comments and thanks for taking into account these concerns.

Answer. We thank the reviewer for their positive responses and for having appreciated our efforts.

References

1. Guenther, F. H. *et al.* A Wireless Brain-Machine Interface for Real-Time Speech Synthesis. *PLoS One* **4**, e8218 (2009).
2. Metzger, S. L. *et al.* A high-performance neuroprosthesis for speech decoding and avatar control. *Nature* **620**, 1037–1046 (2023).
3. Metzger, S. L. *et al.* Generalizable spelling using a speech neuroprosthesis in an individual with severe limb and vocal paralysis. *Nat. Commun.* **13**, 6510 (2022).
4. Moses, D. A. *et al.* Neuroprosthesis for Decoding Speech in a Paralyzed Person with Anarthria. *N. Engl. J. Med.* **385**, 217–227 (2021).
5. Wilson, G. H. *et al.* Decoding spoken English from intracortical electrode arrays in dorsal precentral gyrus. *J. Neural Eng.* **17**, 066007 (2020).
6. Willett, F. R. *et al.* A high-performance speech neuroprosthesis. *Nature* **620**, 1031–1036 (2023).
7. Card, N. S. *et al.* An accurate and rapidly calibrating speech neuroprosthesis. *N. Engl. J. Med.* **391**, 609–618 (2024).
8. Cooney, C., Folli, R. & Coyle, D. Neurolinguistics Research Advancing Development of a Direct-Speech Brain-Computer Interface. *iScience* **8**, 103–125 (2018).
9. Martin, S., Iturrate, I., Millán, J. del R., Knight, R. T. & Pasley, B. N. Decoding Inner Speech Using Electrocorticography: Progress and Challenges Toward a Speech Prosthesis. *Front. Neurosci.* **12**, 1–10 (2018).
10. Alderson-Day, B. & Fernyhough, C. Inner speech: Development, cognitive functions, phenomenology, and neurobiology. *Psychol. Bull.* **141**, 931–965 (2015).

We would like to thank again the Editor and Reviewers for evaluating the revised manuscript. We appreciate the positive feedback and the additional comments. In the point-by-point responses below, we have addressed all the reviewers' comments. The **original comments from the reviewers** appear in bold, *text from the previous version* of the manuscript is in italics, and *newly added text* is in blue italics.

REVIEWERS' COMMENTS:

Reviewer #1 (Remarks to the Author):

I checked the rebuttal letter and the manuscript and for the three reviewers; my impression is that the authors carefully responded and changed accordingly. My own review is not in the list you sent me, my be they responded already. In general I regard this manuscript not as of high priority: the authors conceptualize their experiment as relevant for BCI-training in patients with communication disorders. However, the article ignores completely discussing or even mentioning that a substantial amount of literature at least of locked-in patients has completely failed to generate any verbal communication with EEG-BCI except for "yes" and "no" signals and only invasive recordings allowed verbal communication (see Chaudhary et al Nature Commun. 2022 and others before). Thus the approached here presents some interesting theoretical results but seems to be irrelevant for any future clinical applications of EEG-BCI, in contrast of what they emphasize several times in the manuscript.

However, from a formal point of view, the tone of the three reviews you sent me is positive.

Answer. Thank you for agreeing to review our revision and for bringing up this issue of the relevance of our study to locked-in patients. We fully agree with the Reviewer's comment regarding the limitations of non-invasive EEG for communication purposes and, accordingly, we already wrote at the end of the discussion section (page 18, lines 718-719): "*Although surface EEG-based BCIs are unlikely to become stand-alone communication devices [...]*". We have now expanded this point by adding: "*Training to perform real-time control could be used as a benchmark to select those patients identify learners who would benefit most from invasive intracortical BCIs for communication (Chaudhary et al. 2022)*" (page 18, lines 719-721). In so doing, we now cite the study by Chaudhary and colleagues, which we did not previously include in the discussion due to significant experimental differences (e.g., auditory feedback consisting in the modulation of the frequency of a tone, no imagery or attempted speech as a cognitive task to control the feedback).

Regarding the potential difficulty of using BCI in the Locked-in Syndrome population we previously wrote in the conclusions (page 18, lines 725-726): "*BCIs based on imagined speech might not be suitable for all patients in the long term, such as those in which motor impairments are accompanied by progressive cognitive decline (e.g. LIS, Séguin et al. 2024)*". We had also emphasized the need to gather additional evidence to support the use of such systems in patients (page 18, lines 721-723): "*To validate such an approach, future work is required to 1- establish the correspondence between surface and intracranial EEG recordings of the neural activity elicited by imagined speech*".

Regarding the observation that real-time classification with EEG-BCI is currently limited to "yes" vs "no," this was already mentioned in the introduction as follows (page 3, lines 84-84): "*In the single BCI study that used EEG for online (i.e. closed-loop, real-time) speech imagery decoding (Sereshkeh et al., 2017), performance remained below 70% in discriminating between "yes" and "no".*"

We hope these remarks address the Reviewer's concerns.

Reviewer #3 (Remarks to the Author):

I have reviewed the answers to Reviewer 2. I think the authors responded appropriately, always providing new information and specially clarifying the problem mentioned in point 2, regarding the instructions in the imaginary tasks, which indeed, there was not concrete instructions (this has been now clarified in the text).

Overall, considering all the revision process and current answers, the quality of the paper increased a lot, being the whole review process very constructive.

Answer. We would like to warmly thank the Reviewer for taking on the revision for Reviewer 2; we greatly appreciate this effort. Thank you as well for the positive feedback.

References

Chaudhary, U. *et al.* Spelling interface using intracortical signals in a completely locked-in patient enabled via auditory neurofeedback training. *Nat. Commun.* **13**, 1–9 (2022).

Séguin, P. *et al.* The challenge of controlling an auditory BCI in the case of severe motor disability. *J. Neuroeng. Rehabil.* **21**, 9 (2024).

Sereshkeh, A. R., Trott, R., Bricout, A. & Chau, T. Online EEG Classification of Covert Speech for Brain–Computer Interfacing. *Int. J. Neural Syst.* **27**, 1750033 (2017).